# The Importance of Dietary Antioxidants on Oxidative Stress, Meat and Milk Production, and Their Preservative Aspects in Farm Animals: Antioxidant Action, Animal Health, and Product Quality—Invited Review

**DOI:** 10.3390/ani12233279

**Published:** 2022-11-24

**Authors:** Eric N. Ponnampalam, Ali Kiani, Sarusha Santhiravel, Benjamin W. B. Holman, Charlotte Lauridsen, Frank R. Dunshea

**Affiliations:** 1Animal Production Sciences, Agriculture Victoria Research, Department of Jobs, Precincts and Regions, Bundoora, VIC 3083, Australia; 2Department of Animal Sciences, Faculty of Agriculture and Natural Resources, Lorestan University, Khorramabad P.O. Box 465, Iran; 3Department of Biochemistry, Memorial University of Newfoundland, St. John’s, NL A1C 5S7, Canada; 4Wagga Wagga Agricultural Institute, NSW Department of Primary Industries, Wagga Wagga, NSW 2650, Australia; 5Department of Animal and Veterinary Sciences, Aarhus University, P.O. Box 50, DK-8830 Tjele, Denmark; 6Faculty of Veterinary and Agricultural Sciences, The University of Melbourne, Parkville, VIC 3010, Australia; 7The Faculty of Biological Sciences, The University of Leeds, Leeds LS2 9JT, UK

**Keywords:** oxidation, reactive oxygen species, livestock production, meat and milk quality, health and well-being, antioxidant defense, prooxidants, dietary intervention, bioactive compounds, redox balance

## Abstract

**Simple Summary:**

Oxidative stress occurs within biological systems when free radicals, classified as reactive oxygen species and reactive nitrogen species, are in excess. The reactions between free radicals and other micro-macro-molecules have some benefit but more generally, have been associated with subclinical disease development and suppressed farm animal performances. Subsequently, this may lead to animal-sourced food products having lower quality and shorter shelf-life (off-farm, e.g., meat and milk products). Free radicals are generated from regulated and unregulated sources (e.g., mitochondrial respiration, enzymatic activities, electron leakage), and their responsiveness to environmental and nutritional stressor events overwhelms the antioxidant defenses in the body maintain redox homeostasis, thus leading to oxidative distress. Research has demonstrated that dietary antioxidants may be used to redress redox imbalance and that, in this function, bioactive compounds will offer different efficacies and effects both on- and off-farm. Important dietary antioxidants include vitamins, trace elements, some fatty acids, and phytonutrients, such as beta-carotene, polyphenols, flavonoids, etc. These may be sourced via the feeds, supplements, and forages provided to farm animals, available from natural or synthetic (additive) sources. Antioxidant accumulation within body tissues of farm animals can support better animal performance and improved meat and milk quality with extended shelf-life. Once consumed, these enriched products may help ameliorate human oxidative stress. These were the topics of this review below.

**Abstract:**

The biological effects of oxidative stress and associated free radicals on farm animal performance, productivity, and product quality may be managed via dietary interventions—specifically, the provision of feeds, supplements, and forages rich in antioxidants. To optimize this approach, it is important first to understand the development of free radicals and their contributions to oxidative stress in tissue systems of farm animals or the human body. The interactions between prooxidants and antioxidants will impact redox homeostasis and, therefore, the well-being of farm animals. The impact of free radical formation on the oxidation of lipids, proteins, DNA, and biologically important macromolecules will likewise impact animal performance, meat and milk quality, nutritional value, and longevity. Dietary antioxidants, endogenous antioxidants, and metal-binding proteins contribute to the ‘antioxidant defenses’ that control free radical formation within the biological systems. Different bioactive compounds of varying antioxidant potential and bio-accessibility may be sourced from tailored feeding systems. Informed and successful provision of dietary antioxidants can help alleviate oxidative stress. However, knowledge pertaining to farm animals, their unique biological systems, and the applications of novel feeds, specialized forages, bioactive compounds, etc., must be established. This review summarized current research to direct future studies towards more effective controls for free radical formation/oxidative stress in farm animals so that productivity and quality of meat and milk can be optimized.

## 1. Introduction

Living organisms (flora and fauna) are dependent on nutrient (macro- and micronutrients) and non-nutrient (water, cutin, lignin, pigments, and waxes) components from food to support their growth, development, and proliferation. Macronutrients include carbohydrates, lipids, and proteins composed of carbon, hydrogen, oxygen, nitrogen, phosphorus, sulfur, and other trace elements. These, along with micronutrients (minerals, vitamins, and bioactive compounds), are vital components of the cell and support a broad array of functions in the cellular systems of living organisms. All of these components are essential for normal metabolic activities (anabolism and catabolism) to take place in an organism; most notably in energy production via a chain of oxidation reactions that occur in the tissue compartments and via the utilization of oxygen molecules (O_2_) by cellular systems [1]. Oxidation processes not only provide energy, but they also help in cellular defense actions that are necessary for the maintenance of natural life by producing components that are derived from oxygen, collectively termed as ‘free radicals’. Reactive oxygen species (ROS) and reactive nitrogen species (RNS), derived from oxygen and nitrogen, respectively, are the most common groups of free radicals in biological systems. The production of these compounds at low to moderate levels is expected in farm animals as they are exposed to challenges under extensive or rangeland production systems. However, these oxidative radicals can be harmful to the body at higher levels. These radicals can damage biological macromolecules and form primary and secondary compounds that can multiply the formation of radicals and are destructive to the cellular or tissue systems and their metabolic activities [2].

Taking humans as an example, ≥5% of the inhaled molecular oxygen is converted, by univalent reduction, to ROS, namely superoxide (O_2_^−●^), hydrogen peroxide (H_2_O_2_), and hydroxyl (OH^●^) radicals [3,4]. Thus, all cells and tissues (e.g., gut, muscles, and adipose) under aerobic conditions are under consistent threat from the abuse of ROS. Stressful situations can amplify oxidative processes, thus producing high levels of ROS and causing the associated adverse effects in biological systems [5]. Fortunately, tissues and cells are efficiently protected against free radical damage by a potent defense system called the ‘antioxidant defense system’. This defense system includes exogenous and endogenous components. Dietary antioxidants present in plant foods (e.g., vitamins, phytochemicals, trace elements) and animal foods may be classified as exogenous components as they are ingested as food (feed) and enter into the biological systems via digestion and absorption processes. Endogenous antioxidants include enzymes (e.g., superoxide dismutase, catalase, glutathione peroxidase, glutathione reductase) and non-enzyme components, such as nutrient-based antioxidants (e.g., tocopherols and tocotrienols, carotenoids, and lipoic acid) and metal-binding proteins (e.g., albumin, ferritin, ceruloplasmin, and lactoferrin). Despite the sophisticated antioxidant defense system, under some conditions, the balance between the production of ROS and the antioxidant defense system is weakened or lost, causing a state called ‘oxidative stress’ in the body. Recently, the definition of oxidative stress has been updated to include two aspects, i.e., (1) oxidative eustress, which encompasses a minimum amount of oxidant production essential for life process through redox signaling; and (2) oxidative distress, which encompasses an overexposure to oxidants resulting in non-specific oxidation of biomolecules and disruption of redox signaling [6].

One of the typical situations that is encountered by farm animals is mal- or under-nutrition; often due to prolonged drought or unfavorable seasonal conditions (heat or cold) that impact their daily food consumption and nutrient intakes viz. a balanced diet. This may cause lower growth or poorer health and by doing so, may lead to oxidative distress, as illustrated for grazing cattle and sheep (Figure 1). However, outdoor and/or organic production systems for pig and poultry production may face the same challenges depending on seasonal variations. Oxidative stress deregulates cellular or muscular functions via various events that allow the body to go through various physiological or pathological changes affecting energy metabolism and nutrient partition in the body. This may, in turn, result in reductions in animal performance and product quality. For example, oxidative distress has been proven to negatively impact somatic cell counts in milk [7,8] as well as many different preservatives and sensory aspects of meat [9]. In weaned piglets and broiler chickens, oxidative distress can lead to bacterial infections in the gut, leading to inflammation and poor feed efficiencies due to impaired gut function [5].

Optimal productivity and economic gain from animal production systems are entirely dependent on the well-being and performance of the whole animal flock or herd on the farm and the preservative status of the animal products, such as meat and milk off-farm, from production to the point of consumption (paddock to plate). Apart from the dietary energy and nutrients provided by carbohydrates, fats, and proteins, the natural and artificial (synthetic) antioxidants present in an animal’s diet will play a crucial function in maintaining animal health and well-being. Dietary antioxidants may enhance the performance and productivity of animals (on-farm production), as well as support improvement to the functional aspects (off-farm preservation) of meat and milk [10,11,12,13]. For example, it has been stated that for the dietary vitamin E concentration to be adequate, it should deliver a vitamin E concentration of 3.0 to 3.5 mg per kg of meat as this level is sufficient to maintain the retail color stability of beef [14], sheep meat [15], pork, and poultry meat [16] under commercial retail display. These recommendations are founded on the capacity of α-tocopherol (vitamin E) within muscle tissue systems to promote antioxidant capacities and help to scavenge free radicals, thereby avoiding the formation of oxidative radicals. In doing so, α-tocopherol helps maintain healthy muscle systems via its incorporation in-between the fatty acids of the cellular membranes, thereby preserving the meat, against oxidative deterioration, after slaughter [17,18]. While dietary vitamin E requires time to be absorbed and incorporated into muscle tissue, vitamin C (ascorbic acid), which is a potent water-soluble antioxidant able to scavenge oxidants and to regenerate other antioxidants, may, after on-farm supplementation close to slaughter, improve meat quality. However, the impact seems dependent on the timing relative to slaughter.

The knowledge of the importance of oxygen-derived prooxidants and antioxidants in both normal metabolism and the status of various subclinical diseases has increased recently. Antioxidants may act as prooxidants depending on the specific productive status of the animal, metabolic demand, and oxidative stress conditions. Particularly, their dosage levels and the redox potential of the cell are important factors. Even though antioxidants may help to reduce free-radical-induced oxidative damage in the cells or tissues, leading to diminished toxicity, human studies have shown that the non-oxidative cytotoxic mechanisms of specific drugs may not be influenced by antioxidant supplementation [19].

There are many important aspects of antioxidants from feed (food) in maintaining farm animal health, meat and milk productivity, the preservation of animal product quality, and functional features for better human health (Figure 2), which are not fully explored. This review aimed to provide a detailed understanding of the sources of free radical formation, mainly ROS, their consequences and oxidative stress in the biological systems of farm animals, and the importance of dietary antioxidant action to maintaining animal performance and the functional aspects of meat and milk, namely as nutrient-enriched food that is safe for human consumption.

## 2. Development of Reactive Oxygen Species and Oxidative Stress in Living Organisms

### 2.1. Development of Reactive Oxygen Species

ROS and RNS are terms that encompass all more reactive, oxygen-containing compounds, including free radicals. ROS is also termed reactive oxygen intermediates, active oxygen species, or reactive oxygen derivatives. Types of ROS include the oxygen-radicals, such as superoxide anion (O_2_^−●^), hydroxyl radical (OH^●^), radical alkoxyl (RO^●^), perhydroxyl radical (HO_2_^●^), hydrogen peroxide (H_2_O_2_), and radical peroxyl (ROO^●^). RNS include nitric oxide radical (NO or NO^●^), nitrite (NO_2_^−^), nitrogen dioxide radical (NO_2_^●^), and peroxynitrite (ONOO^−^) [20]. ROS are generally produced by tightly regulated enzymes, namely endothelial NO synthase [21] and NAD(P)H oxidase isoforms, xanthine oxidase, and cytochrome P450 enzymes. ROS and RNS play a dual role with both beneficial and harmful aspects. The positive effects of ROS (particularly, superoxide radical) occur when ROS is produced at low or moderate concentrations within biological systems as they are involved in many physiological roles, such as in cellular responses and maintenance. ROS is also involved in the defense mechanism towards infectious agents, the stimulation of a mitogenic response, and the function of several cellular signaling pathways [22].

Paradoxically, many ROS-mediated actions protect cells against ROS-induced changes and maintain ‘redox balance’—also called ‘redox homeostasis’. The two-way characteristics of ROS are clearly shown in research findings. In this context, a growing body of research demonstrates that ROS, within cells or tissue systems, act as a secondary messenger in intracellular signaling cascades, which stimulate and maintain the oncogenic phenotype of cancer cells. Nevertheless, ROS can also function as anti-tumorigenic species as they induce cellular senescence and apoptosis.

Oxygen derivatives, particularly superoxide anions and hydroxyl radicals, are the chief free radicals in several disease conditions. Both environmental and endogenous factors induce the formation of radicals in the body via several mechanisms. Superoxide anions are formed by transferring a single electron to oxygen, and various mechanisms are involved in the in vivo production of superoxides. Transition metals, such as iron and copper, greatly accelerate the oxidation of some molecules, such as flavine nucleotides and thiol compounds, in the presence of oxygen to generate superoxide. The electron transport chain in the inner mitochondrial membrane is responsible for reducing oxygen to water. Free radical intermediates produced during this process are tightly bound to the components of the transport chain. However, there is a constant leak of a few electrons into the mitochondrial matrix, which results in superoxide formation [23]. There may also be continuous generation of superoxide by other cells to regulate cell differentiation and growth; generation of superoxide anions by the vascular endothelium to neutralize nitric oxide; and the generation of superoxide by phagocytic cells during the oxidative burst [19].

Any biological system producing hydrogen peroxide will also produce superoxide anions because of a spontaneous dismutation reaction (2 O_2_^●−^ + 2 H^+^→ H_2_O_2_ + O_2_). Moreover, some enzymatic reactions may directly produce hydrogen peroxide. Hydrogen peroxide is not a free radical as it does not contain any unpaired electrons [24]. However, it is a precursor to certain radical species such as hydroxyl radical, peroxyl radical, and superoxide. The key characteristic of hydrogen peroxide is its ability to freely permeate cell membranes, which superoxide generally cannot. Therefore, hydrogen peroxide produced in one location might travel a significant distance across the cells before decomposing to generate the highly reactive hydroxyl radical, likely to mediate most of the toxic effects attributed to hydrogen peroxide. Hydrogen peroxide functions as a channel to transmit ‘free radical-induced damage’ across cell compartments and between cells.

In animals, during normal metabolic processes, ROS are generated via several mechanisms within cellular or tissue compartments, including the mitochondria, chloroplast, endoplasmic reticulum, peroxisomes, apoplast, plasma membrane, and cytoplasm, and are as such the part of the oxidative eustress, i.e., a minimum amount of oxidant production essential for life processes [6]. They can, however, also be formed as a result of stimulation by environmental agitations, such as nutritional deficiency, drought, salinity, solar heat and radiation, exposure to heavy metals, and exposure to pesticides and herbicides [25], hence leading to oxidative distress, i.e., overexposure to oxidants. ROS development in any living organism is a sequential process. In the series of univalent processes by which O_2_ undergoes reduction, numerous reactive intermediates are produced, such as superoxide (O_2_^−●^), hydrogen peroxide (H_2_O_2_), and the highly reactive hydroxy radical (OH^●^) named as the ROS through the process shown below:O_2_ → O_2_^−●^ → H_2_O_2_ → OH^●^ → H_2_O

Finally, the presence of a trace amount of H_2_O_2_ and Fe^2+^ salt is required for the in vivo generation of OH^●−^ except times of exposure to radiation, leading to the Fenton reaction shown below:Fe^2+^ + H_2_O_2_ → Fe^3+^ + OH^●^ + OH^−^

At extreme reactive conditions, such as high concentration, ROS can result in non-controlled oxidation in the cells and tissues of biological systems, known as oxidative stress. The attack of ROS on the cellular or tissue (e.g., muscular in animals) compartments of plants and animals can damage membrane lipids, proteins, nucleic acids, enzymes, and other small molecules on the organelle or body, resulting in extensive cellular and tissue damage [1,26]. These reactions can modify intrinsic membrane properties like fluidity, ion transport, loss of enzyme activity, membrane signal transduction, DNA damage, and protein synthesis. At extreme levels, they can cause cell or tissue necrosis (Figure 3). These damages to cells and tissues, caused by free radicals, are believed to play a major role in developing various metabolic disorders in animals and plants. Further, in farm animals, it can lead to the development of subclinical diseases, leading to poor animal performance and productivity, ultimately leading to the quality deterioration of milk and meat products [10,27].

### 2.2. Development of Oxidative Stress

Oxidative (di)stress is a natural phenomenon of aerobic organisms. It encompasses endogenous and exogenous factors. Oxidative stress results from disequilibrium between oxidants and antioxidants, which in turn is revealed by a continuous increase in the production of ROS (Figure 4). ROS comprise radicals and other reactive oxygen factors that can react with other substrates—some examples of ROS in biological systems have been mentioned earlier. Under physiological conditions, these are counterbalanced by numerous defense pathways. It should be emphasized that ROS have many beneficial physiological roles; for example, in cell signaling, a defense against infectious agents, and maintenance of good health in the natural body [1]. However, when produced in excess or in circumstances where defense systems are compromised, ROS may react with macromolecules in cellular or tissue compartments, namely proteins, fatty acids, DNA, and to some extent, carbohydrates, thereby causing further damage to these substrates and leading to the development of a state called oxidative stress [10,19].

There has been considerable literature on oxidative stress and the development of metabolic disorders in humans and farm animals. Oxidative stress affects both humans and farm animals for relatively similar reasons. Humans and livestock maintain a wide range of antioxidant defense systems that help to alleviate or fight against oxidative stress. Many researchers stated that oxidative stress-related mechanisms are important in developing several disorders in farm animals. If not remedied, oxidative stress can be harmful and cause metabolic disorders in animals that may lead to lower production [28].

A multi-dynamic antioxidant defense system is present in living organisms to protect the cells and tissues from damage caused by oxidative stress. This system encompasses non-enzymatic antioxidants (e.g., vitamins, polyphenols (flavonoids), carotenoids), metal binding proteins (e.g., ferritin, albumin), and enzymatic antioxidants (e.g., glutathione peroxidase (GPX), superoxide dismutase (SOD), and catalase) [29,30]. It is believed that these enzymatic and non-enzymatic antioxidants, found in the biological (circulatory and tissue) systems, can break down the chains of free radical formations by directly scavenging the free radical forming agents (e.g., H_2_O_2_) or chelate the metal ions that act as an oxidative base (e.g., Fe^2+^). The latter process may act on removing ROS in the system or the deactivation of further reactions of ROS in the body [31]. This will protect protein, lipid, and DNA in the cellular systems from oxidative stress damage and improve immune responses, which may increase animal performance and productivity [32,33,34].

#### 2.2.1. Oxidative Stress and Mitochondrial Damage

Mitochondria play a significant role in several cellular functions that include energy production, respiration, metabolism of amino acids, nucleotides, and iron, synthesis of heme and lipids, maintenance of intracellular homeostasis of inorganic ions, cell motility, cell proliferation, and apoptosis. Mitochondria contain their own DNA (mtDNA), and mtDNA occurs in small clusters called nucleoids or chondrodites. In response to physiological conditions, the number of mtDNA molecules in nucleoids will differ in number and size. The majority of the mitochondrial proteins are encoded by the nuclear DNA, while mitochondrial DNA encodes only a few of these proteins. mtDNA is more vulnerable to oxidative damage compared to nuclear DNA, mainly for three reasons: (1) its proximity to the electron transport chain; (2) its continuous exposure to ROS that is produced during oxidative phosphorylation; and (3) its limited capacity or strategies for DNA repair and the lack of protection by histones [35]. Although various cellular systems (xanthine oxidase, NADPH oxidase, cytochrome P450 enzymes, and endothelial NO synthase) can generate ROS, mitochondria are the principal organelles for ROS formation in most mammalian cells. The generation of mitochondrial ROS results from oxidative phosphorylation at the respiratory chain complexes, where the electrons obtained from NADH and FADH_2_ can directly react with oxygen or other electron acceptors and produce free radicals [36,37]. Mitochondria are also a major site for accumulating low molecular weight Fe^2+^ complexes, which promote the oxidative damage of membrane lipids [38]. Mitochondria not only represent the primary source of ROS, but they are also the major targets of its damaging effects. Over 20 types of mutagenic base modification are produced in DNA by ROS and these DNA lesions may cause mutations in mtDNA that can impair mitochondrial function [35].

Mitochondrial turnover is important for normal energy production, maintenance of a healthy mitochondrial phenotype, and the promotion of healthy life because macromolecules in mitochondria (including mtDNA) are particularly susceptible to oxidative damage. Mitochondria are important cellular targets of insulin-like growth factor-I (IGF-I). Previous research has shown that mitochondrial superoxide generation is reduced by IGF-I [39] and increased oxidative stress damage is associated with the low levels of IGF-I [40]. Mitochondria are highly dynamic organelles, and deregulation of mitochondrial turnover is likely to be one of the intrinsic causes of mitochondrial dysfunction, which lead to the deregulation of cell metabolism, oxidative stress, and altered signal transduction during the ageing process. In poultry, evidence strongly supports that heat stress induces oxidative stress. Although heat (exposure) caused oxidative stress can manifest in all parts of the body, mitochondrial dysfunction that underlies oxidative stress has recently received much interest [41]. The importance of heat stress in swine production is less studied compared to poultry production. However, it has been found that acute heat stress in pigs can contribute to oxidative damage [42]. Additionally, the genetic background can influence the oxidative stress level in pigs, as breeding towards lean-type pigs has resulted in chronic oxidative stress conditions causing poor homeostatic control of inflammatory responses [43].

#### 2.2.2. Oxidative Stress and Antioxidant Defence

Under natural conditions, antioxidant defenses regulate ROS generation via the enzymatic and non-enzymatic defense systems of the body. In farm animals, as shown in Table 1, nutritional deficiencies or imbalance in dietary nutrients can cause oxidative stress and metabolic disorders [10]. High performing animals are often under high nutrient demand, which may enhance the development of free radicals, if the feeding systems or environment is not ideal for their production potential [44]. Therefore, animals with high performance potential are sensitive or prone to oxidative stress. With animals with a subclinical disease level status, the body would also enhance endogenous antioxidant defense mechanisms during immune responses, given that the antioxidant potential of the body is optimal or above the threshold. At optimal antioxidant potential, the immune system will produce defensive substances specifically during infection. This mechanism can deregulate the functions of the infectious agent and block the infection naturally. Therefore, diets rich in antioxidants are essential for maximum farm productivity. However, during impaired antioxidant status, such as the newly weaned piglet, which often suffers from reduced vitamin E and vitamin C status, oxidative distress may be induced. This is because during the weaning period, the risk of enteric infection is high due to the pathogenic exposure of a still immunologically immature piglet. Supplementation of the host’s capacity to detoxify ROS via dietary strategies is the only identified mechanism by which heat-induced oxidative stress in poultry can be limited [41]. Thus, supplementation with antioxidative vitamins and trace elements controls oxidative stress and thereby prevent excessive production of ROS during a host-inflammatory induced response. This, in turn, can help to prevent enteric infection [45] or as shown for poultry, antioxidant phytochemicals have shown positive results under challenging conditions of heat-induced oxidative stress [41].

Recently, the supplementation of synthetic antioxidants as feed additives has been discouraged in food preservation and/or animal-based food production [46]. For example, there are limits on the concentrations of selenium, a potent antioxidant, that can be added to the diet in many parts of the world due to concerns about toxicity. While these concerns are really only an issue for inorganic forms of selenium, there are blanket limits on all forms of selenium. Due to these reasons, researchers around the world are searching for antioxidants of natural origin or feeds containing natural antioxidants for animal production on farm and post farm food preservation [47,48]. In the last two decades, there has been remarkable changes in the development of biomarkers of oxidative stress and prevention of diseases in human. However, there are still challenges associated with livestock production and their health relating to antioxidant status because changes in climate, agriculture practices, and feeding habit all impact farm animal production. In this regard, more investigations are needed to (1) assess the use of dietary antioxidants on oxidative stress, animal health, and product quality using well-designed studies in farm animals; (2) examine the basal levels of oxidative stress associated with animal behavior, metabolic status, and feeding habits; and (3) validate available biomarkers for the assessment of oxidative stress in farm animals based on their dietary antioxidant intake and production purpose. It is reasonable to state that the responses from an individual animal on oxidative stress and productivity loss can differ depending on the antioxidant intake (as feeds or supplements), tissue antioxidant status, and amount of ROS produced.

It should be noted that antioxidant application at high doses may be deleterious. Some negative impacts of antioxidants when used in dietary supplements, such as selenium, vitamins, flavonoids, carotenoids, α-lipoic acid, and synthetic compounds, have been identified [49]. For example, depending on the dose, ascorbic acid exhibits both prooxidant and antioxidant properties. Ascorbic acid is classified as an antioxidant due to the low electron potential and resonance stability of ascorbate and the ascorbyl radical. In ascorbic acid treated rats, ascorbic acid can act as a cytochrome P450 inhibitor [50]. Similar activity has also been observed for other antioxidants viz. quercetin and chitosan oligosaccharides, which may act as potential CYP inhibitors. The in vivo prooxidant/antioxidant activity of beta-carotene and lycopene is dependent on their interaction with biological membranes and the other co-antioxidant molecules like vitamin C or E. Carotenoids tend to lose their effectiveness as antioxidants at higher oxygen tension. Paradoxically, the prooxidant effect of low levels of tocopherol is evident at low oxygen tension, a topic covered in more detail by Rahal et al. [49] and Kurutas [19].

With regard to human diseases (or pathologies), antioxidant supplementation or antioxidant therapy have no clear effect on the risk of chronic diseases such as cancer, diabetes mellitus, cardiovascular disease, coronary artery disease, and neurodegenerative diseases. The successful development of effective antioxidant therapies remains a vital point to explain the role played by the accumulation of oxidized molecules in disease development, relating to oxidative stress. Kurutas [19] stated that any human intervention trial that does take place should be accompanied by measurements of one or more relevant biomarkers at intervals during the study. If the endpoint of the trial is disease incidence or mortality, such studies could help to validate or disprove the biomarker concept. In this context, the antioxidant (therapy) action concept on curing health threatening human disease is rather different from the antioxidant action on enhanced animal performance and product quality in farm animals. Therefore, the potential of antioxidants (actions) as feed additives on animal performance and product quality of meat and milk in farm animals should be taken in different perspective or category, when compared with the potential of antioxidants (actions) as a drug therapy on treating heath threatening human diseases.

## 3. Sources of Reactive Oxygen Species and Free Radicals in Biological Systems

All aerobic organisms produce ROS. There are many productive pathways in the cells and the whole organism that are involved in regulating the production of ROS and the follow-on effects on signaling cascades. The generation of ROS in the biological systems can be due to the non-regulated production of ROS, regulated production of ROS, and other cellular ROS sources as shown in Figure 5. The non-regulated modes are given below. However, there are many sources within the cells that are only mentioned.

### 3.1. Non-Regulated Production of Reactive Oxygen Species

Several systems have been found to be involved in ROS production and pathologies caused by the ROS. These include nicotinamide adenine dinucleotide phosphate oxidases (NOXs), oxidoreductases (e.g., xanthine oxidase), uncoupled nitric oxide synthase, and enzymes and cofactors present in mitochondria, such as cytochrome b5 and cytochrome c [51,52].

#### 3.1.1. Mitochondrial Respiration

Mitochondrial respiration is the major source of endogenous ROS production (90% of cellular ROS) due to its role in oxidative ATP production, in which molecular oxygen (O_2_) is reduced to water. The mitochondrial electron transport chain (ETC) encompasses adequately energized electrons to reduce O_2_ to generate ROS. In fact, 0.2–2.0% of the molecular oxygen consumed by the mitochondria is reduced to superoxide anions [48]. However, overproduction of ROS by mitochondria may occur during infectious disease in animals such as observed in poultry [41]. The superoxide radical (O_2_^•−^) is produced at a number of sites in the mitochondria, including complex I (sites IQ and IF), complex III (site IIIQo), glycerol 3-phosphate dehydrogenase, Q oxidoreductase, pyruvate dehydrogenase, and 2-oxoglutarate dehydrogenase [53]. Still, two major components in the ETC, called complex I (NADH dehydrogenase) and complex III (ubiquinone cytochrome c reductase), largely contribute for the formation of superoxide radicals [50].

The majority of the mitochondrial ROS are generated during the oxidative phosphorylation process in the ETC, where O_2_ is reduced to H_2_O. Complex I transfers two electrons from nicotine adenine dinucleotide (NADH) to ubiquinone (Q) and pumps four protons into the intermembrane space [51]. The NADH, produced by the tricarboxylic acid cycle in the mitochondrial matrix, donates two electrons to a Flavin mononucleotide (FMN) located at the complex I to generate FMNH_2_ [52]. Then, these electrons pass to coenzyme Q or ubiquinone (Q) to form the reduced form of ubiquinone (QH_2_) at the Q binding site. An unstable intermediate semiquinone anion (^•^Q^−^) in the Q-cycle can regenerate ubiquinone from the reduced QH_2_. The generated ^•^Q^−^ transfers electrons to O_2_ producing superoxide radicals (O_2_^−•^). During the electron transfer, electrons leak and interact with O_2_ to form superoxides at the FMN site. In the complex I, ROS are produced at both the FMN site and Q binding site [53]. Besides, ROS production in the complex I is further enhanced by the reverse electron transport from NAD+ and QH_2_ [54].

In complex III, the fully reduced form of ubiquinone donates an electron to Cytochrome c1. The unstable ubisemiquinone, a primary electron donor which is produced as an intermediate during the Q cycle, has the ability of reducing O_2_ to superoxides [55]. During the Q cycle, ubisemiquinone of the outer Q site moves freely in complex III and leaks a single electron to O_2_ forming ROS. The produced superoxides are released into the matrix and intermembrane space by the Qi ad Qo sites, respectively [56]. In addition to complexes I and III, the FAD site of complex II also can produce less number of superoxides towards the matrix. Both forward and backward reactions in the complex II contribute to the ROS production using electrons derived from succinate and reduced ubiquinones, respectively [57]. Other than ETC, there are several enzymes in the mitochondrial matrix that are capable of generating ROS. For instance, mitochondrial glycerol-3-phosphate dehydrogenase produces ROS (superoxides) by oxidizing glycerol-3-phosphate and reducing Q to QH_2_ [58]. Moreover, 1-Galactono-γ-lactone dehydrogenase indirectly generates ROS by supplying electrons to the ETC and aconitase directly produces ROS [59]. Other enzymes namely flavoprotein—ubiquinone oxidoreductase, 2-oxoglutarate dehydrogenase, dihydrolipoamide dehydrogenase, pyruvate dehydrogenase, and dihydroorotate dehydrogenase also contribute for the mitochondrial ROS generation [48].

The superoxide radicals are converted to hydrogen peroxide (H_2_O_2_) by superoxide dismutase in the mitochondrial membrane or cytosol. It was found that 1–5% of the total O_2_ consumed by the mitochondria is utilized for the generation of H_2_O_2_. The H_2_O_2_ can further be converted by mitochondrial aconitase to a hydroxyl radical (^•^OH) via a Fenton reaction [54].

One more site of ROS production in the mitochondria is the cytochrome catalytic cycle. The cytochrome P450 (CYP) enzymes are a various group of heme monooxygenases that, through the course of their reaction cycle, contribute to cellular reactive oxygen species [55]. Cytochrome enzymes metabolize a wide range of organic substrates which give rise to superoxide radical and H_2_O_2_ as by-products [56]. Cytochrome P450 oxidase is part of the microsomal electron transport system. It belongs to the CYP superfamily of integral membrane proteins that catalyze the oxidation of numerous organic substrates, accompanied by the reduction of molecular oxygen [55]. CYP enzymes also have peroxygenase and peroxidase activity, using H_2_O_2_ either for direct oxidation of substrates or as a donor of oxygen atoms [57]. H_2_O_2_ and superoxide radicals are produced during the CYP monooxygenase cycle. H_2_O_2_ can be further decomposed to hydroxyl radicals (^•^OH) in the presence of ferrous iron via a Fenton reaction. Figure 6 simply illustrates the generation of ROS in the mitochondria and cellular matrix of living organisms.

#### 3.1.2. Peroxisomes

Peroxisomes are subcellular spherical microbodies covered by a single membrane without DNA. They perform oxidative metabolism and are the major site for intracellular H_2_O_2_ generation. The chief metabolic process that produces H_2_O_2_ in the peroxisome is the β-oxidation of fatty acids [60]. In the respiratory pathway of peroxisomes, electrons are transferred from several metabolites reducing O_2_ to H_2_O_2_. Moreover, the respiratory pathway of peroxisomes is not associated with the oxidative phosphorylation or generating ATP, instead release free energy in the form of heat [61]. Various peroxisome enzymes, photorespiratory glycolate oxidase in green tissues and particularly flavoproteins, including acyl-CoA oxidases, D-aspartate oxidase, xanthine oxidase, D-amino acid oxidase, polyamine oxidase, L-pipecolic acid oxidase, urate oxidase, and L-α-hydroxyacid oxidase (Table 1), produce H_2_O_2_ as a part of their normal catalytic cycle [62]). It was reported that the rate of H_2_O_2_ generation by the peroxisomal glycolate oxidase is nearly 2–50 times greater than that of mitochondria and chloroplast [63].

Peroxisomes are also capable of producing other free radicals such as O_2_^•−^, OH^•^, and NO^•^. Two different sites of peroxisomes, namely NADPH-dependent small ETC and xanthine oxidase in the peroxisomal matrix, generate O_2_^•−^. NADH and Cyt b components of NADPH-dependent small ETC present in the peroxisomal membrane utilize O_2_ as the electron acceptor and release O_2_^•−^ into the cytosol. Xanthine and hypoxanthine in the peroxisomal matrix are metabolized by the xanthine oxidases into uric acid and produce O_2_^•−^ as by-product [62]. Besides, Peroxisomal Membrane Polypeptides (integral membrane proteins) also generate O_2_^•−^, where NADH and NADPH act as electron donors to reduce Cytochrome c. Figure 7 illustrates the process of H_2_O_2_ and O_2_^•−^ generation by the peroxisomes. Moreover, xanthine oxidase is capable of reducing nitrates and nitrites to NO^•^. Fransen et al. [58] proposed that the generation of ONOO^−^ by the peroxisome is kinetically and thermodynamically feasible [62].

#### 3.1.3. Endoplasmic Reticulum

Endoplasmic reticulum enzymes, namely diamine oxidase and cytochrome p-450, (Cyt P450) and b5 enzymes are responsible for the generation of ROS. Cytochrome p-450 involve in the NADPH-mediated electron transport chain in the ER to produce O_2_^•−^ [64]. First, Cyt P450 form a free radical intermediate (Cyt P450 R^−^) by interacting with an organic substrate and are reduced by flavoprotein. This intermediate reacts with triplet O_2_, forming an oxygenated compound (Cyt P450-ROO^−^). Then, the complex is decomposed to Cyt P450-Th and produces O_2_^•−^ as a byproduct [65]). Moreover, thiol oxidase enzyme (Erop1p) of the endoplasmic reticulum catalyzes the electron transfer to dithiols to O_2_ generating H_2_O_2_ [66].

#### 3.1.4. Xanthine Oxidoreductase

Xanthine oxidoreductase (XOR) is a molybdenum iron-sulfur flavin hydroxylase present in all mammalian tissue and fluid, which exists in two forms (1) xanthine dehydrogenase (XDH); and (2) xanthine oxidase (XOD). The enzyme catalyzes the oxidation of hypoxanthine to xanthine or xanthine to uric acid during purine metabolism [59]. Uric acid and its oxidized derivatives may exhibit prooxidant activities. Additionally, XOR oxidizes several xenobiotics and various endogenous metabolites. XOD, but not XDH, generates H_2_O_2_ and O_2_^•−^ through NADH oxidation [67]. XOD transfers an electron directly to O_2_ and generates O_2_^•−^, subsequently producing H_2_O_2_. Later, Haber–Weiss and Fenton reactions form hydroxyl radical (HO^•^) in the presence of iron. However, XDH is capable of producing these ROS under hypoxic conditions and can oxidize NADH at the FAD site. Low O_2_ tension and acidic pH of hypoxia condition increase affinity of XOR for nitrites, which compete with xanthine and could be reduced to NO by XOR [68]. Consequently, XOR-released superoxide radicals react with nitric oxide (NO^•^), generating peroxynitrite (ONOO^−^). NO^•^, in turn, is produced by NOS activity or even by XOR under hypoxic conditions [69].

#### 3.1.5. Dopamine

Dopamine is a neurotransmitter which is produced from dopamine neurons. Dopamine is an unstable molecule that may auto-oxidize to ROS, especially H_2_O_2_ [70]. Oxidative metabolism of dopamine by monoamine oxidase also produces H_2_O_2_ as a by-product. This H_2_O_2_ can subsequently react with O_2_ or iron to generate more reactive HO^•^. Dopamine quinones may react with the sulfhydryl groups of the cysteine amino acid in glutathione, leading to the lower level of glutathione (a ROS scavenger) and higher levels of ROS [71]. Besides the synthesis and degradation, the transport and storage of dopamine also contribute to elevated ROS production. Any perturbation elevating cytoplasmic dopamine can increase dopamine auto-oxidization and subsequently increase ROS production in cells [72].

#### 3.1.6. Photosensitization Reactions

Photosensitization reactions occur in animals due to the increased susceptivity of skin to damage triggered by ultraviolet radiation (UVB and UVA radiations). Additionally, skin exposure to ionizing radiation, xenobiotic, and drugs produce ROS in the body. Photosensitization reaction in the skin is also known as photoaging. It was found that several photosensitizing agents in the skin trigger the generation of ROS, such as singlet oxygen (^1^O_2_), superoxides (O_2_^•−^), and hydroxyl radicals (^•^OH) [73]. Chromophore activates the ground state oxygen to absorb UV radiation and generate ROS because energy of photos (320–400 nm) is not directly transferred to the ground state O_2_. Examples of endogenous chromophores are porphyrins, riboflavin, NADH, NADPH, and quinones [74]. Free radical activated reactions are commonly chain reactions. Activation of ground state oxygen produce ^1^O_2_, O_2_^•−^, and H_2_O_2_ as a result of normal metabolism. Both O_2_^•−^ and H_2_O_2_ subsequently produce highly reactive ^•^OH radicals via iron (Fe^2+^) mediated Fenton and Haber–Weiss reactions [75].

Besides, conversion of arginine to citrulline by nitric oxide synthase produces NO, which reacts with the O_2_^•−^ and generates reactive nitrogen species (RNS), particularly ONOO^−^. Moreover, xenobiotics are converted to toxic quinones by cytochrome P450 enzymes. These quinones are reduced to hydroquinones/semiquinones and produce O_2_^•−^. ROS can interact with the lipid rich plasma membrane and lead to lipid peroxidation reaction [75]. In addition, these ROS and RNS affect the cutaneous collagen and other proteins, causing a number of skin disorders, as well as damages to DNA.

### 3.2. Regulated Production of Reactive Oxygen Species

#### 3.2.1. Nitric Oxide Synthase

Nitric oxide synthases (NOSs) are a family (three isoforms) of enzymes catalyzing the production of nitric oxide (NO) from L-arginine. NOSs produce endogenous NO by converting L-arginine to L-citrulline. NOSs include neuronal NOS (nNOS), endothelial NOS (eNOS), and inducible NOS (iNOS). Calmodulin and Ca^2+^ ions are required for the activation of nNOS and eNOS isomers, while iNOS is greatly independent of Ca^2+^ ions. Peroxynitrite (ONOO^−^) is formed as result of NO scavenging by ROS. The resulting ONOO^−^ not only oxidizes DNA, proteins, and lipids but also interferes with important vascular signaling pathways. Uncoupled eNOS could generate ROS rather than NO. l-arginine is a substrate that NOS utilizes for tetrahydrobiopterin (BH4) binding to the oxygenase domain of eNOS. Then, BH4 stabilizes the dimer as one of the cofactors of NOS activation. In the absence of BH4, eNOS dimer would be uncoupled into two monomers. Uncoupled eNOS is less efficient in the production of NO and generates large amounts of ROS [76]. Depending on the availability of substrate and cofactor, all three isomers of NOS produce O_2_^•−^. NOS isoforms catalyze the reduction of O2 to O_2_^•−^ when the NOS is not saturated with the cofactor BH_4_ [77].

#### 3.2.2. NADPH Oxidase

The NADPH oxidase (NOXs) family are transmembrane proteins that transfer an electron from the NADPH substrate to FAD across biological membranes in order to reduce oxygen to a superoxide radical [78]. The NOXs family includes seven members: NOX1, NOX2, NOX3, NOX4, NOX5, dual oxidase 1 (DUOX1), and DUOX2 [79]. NOXs enzymes reduce molecular oxygen to a superoxide as a primary product, and this is further converted to various ROS [80]. Dysregulation of NOX activity leads to elevated ROS production [81].

#### 3.2.3. Arachidonate Cascade Enzymes

Arachidonate cascade enzymes include cyclooxygenases (COXs) and lipoxygenases (LOXs) and participate in PUFA metabolism. COX and LOX oxygenate arachidonic acid, resulting in the formation of prostaglandin G2 and H2 (PGG2/PGH2) and fatty acid hydroperoxide, respectively, reactions that are accompanied by ROS release [65]. There are several lipoxygenases which differ by substrate specificity and optimum reaction conditions. Lipoxygenases in plants and animals are heme containing dioxygenases that oxidize PUFA at specific carbon sites to give enantiomers of hydroperoxide derivatives with conjugated double bonds. The number in specific enzyme names such as LOX5, LOX12, or LOX15 refers to the arachidonic acid site that is predominantly oxidized [60]. LOX5 is best known for its role in biosynthesis of the leukotrienes A4, B4, C4, D4, and E4. The oxidized metabolites generated by LOX5 were found to change the intracellular redox balance and to induce signal transduction pathways and gene expression. The enzyme LOX5 has been identified as an inducible source of ROS production in lymphocytes [61,62]. LOX5 was shown to be involved in the production of H_2_O_2_ by T lymphocytes after ligation of the CD28 costimulatory receptor [82,83] and in response to interleukin-1β [61]. A lipid metabolizing enzyme in fibroblasts similar to LOX15 has been shown to generate large amounts of extracellular O_2_^−●^ [63].

#### 3.2.4. Cyclooxygenase (COX-1)

Cyclooxygenase (COX), the key enzyme required for the conversion of arachidonic acid to prostaglandins COX, is present in three isoforms: (1) constitutive (COX-1); (2) inducible (COX-2); and (3) splice variant of COX-1 (COX-3). Two cyclooxygenase isoforms have been identified and are referred to as COX-1 and COX-2. Under many circumstances, the COX-1 enzyme is produced constitutively (i.e., gastric mucosa) whereas COX-2 is inducible (i.e., sites of inflammation) [64]. COX-1 has been implicated in ROS production through formation of endoperoxides, which are susceptible to scavenging by some antioxidants in cells stimulated with TNF-α, interleukin-1, bacterial lipopolysaccharide, or the tumor promoter 4-Otetradecanoylphorbol-13-acetate [66]. Cyclooxygenase participation in redox signaling remains scarce.

### 3.3. Other Cellular ROS Sources

The most studied producers of O_2_^−●^ by oxidizing unsaturated fatty acids and xenobiotics are cytochrome P450 and the b5 family of enzymes [63]. Electrons leaking from nuclear membrane cytochrome oxidases and electron transport systems may give rise to ROS [84]. In addition to intracellular membrane-associated oxidases, aldehyde oxidase, dihydroorotate dehydrogenase, flavoprotein dehydrogenase, and tryptophan dioxygenase can all generate ROS during catalytic cycling. pH-dependent cell wall peroxidases, germen-like oxalate oxidases, and amine oxidases have been proposed as a source of H_2_O_2_ in the apoplast of plant cells [85]. Glycolate oxidase, D-amino acid oxidase, urate oxidase, flavin oxidase, L-α-hydroxy acid oxidase, and fatty acyl-CoA oxidase are important sources of total cellular H_2_O_2_ production in peroxisomes [63]. Auto-oxidation of small molecules, such as epinephrine, flavins, and hydroquinone, can also be an important source of intracellular ROS production [68].

## 4. The Impact of Radical Formation on the Oxidation of Biological Macromolecules in Cellular System

Oxidation of lipids, proteins, DNA, and carbohydrates are the major macromolecules for the formation of radicals in biological systems. The formation of the free radical in the biological systems may result in the development of primary and secondary mediators, leading to changes in pathophysiology in the cellular and tissue compartments

### 4.1. The Oxidation of Lipids

Lipids perform many different functions in a cell. Cells store energy for long-term use in the form of lipids. Lipids provide insulation from the environment for plants and animals. Due to their water-repelling nature, fur, skin, and feathers made of wax and cutin lipids help keep aquatic birds and mammals dry. Lipids are also an important constituent of the plasma membrane and are the building blocks of many hormones. Lipids include neutral fats (plant oils and animal fats), wax, phospholipids, cholesterol, and steroids. The oxidation of lipids is believed to be the most damaging process known to occur in biological systems of living organisms, e.g., blood [86] and animal products post-mortem, e.g., muscle tissues or milk. These reactions lead to the development of oxidized lipids and fatty acids that give rise to free radicals. Oxidation products, such as (2E)-4-hydroxyalk-2-enals and aldehydes (malondialdehyde), alkanes, lipid epoxides, and alcohols, can further react with proteins and nucleic acids. The overall effects of lipid oxidation can cause many aspects—decrease in membrane fluidity, an increase in the leakiness of cell substances and damage to membrane proteins, inactivation of receptors, enzymes, and ion channels. Oxidation of lipids can be induced by ROOH, ^3^O_2_, ^1^O_2_, or catalyzed by metal ions (e.g., Cu^2+^ or Fe^2+^).

#### Oxidation of Lipids Caused by ROOH

The ROOH of fatty acids and their radicals may react in three ways. In the first way, there is no change in the number of carbon atoms in the molecule. ROOH species from polyunsaturated fatty acids (PUFAs) containing three or more double bonds in a molecule are unstable. They tend to pass in 1,4 cyclization to the six-member peroxides derived from 1,2-dioxanes, which are also unstable compounds and decompose to low molecular active products. ROOH molecules by 1,3 cyclization pass to five-member peroxides, 1,2-dioxolanes, and endoperoxides. The main malondialdehyde precursors emerge from 1,2 dioxolane-type peroxohydroperoxides. ROOH and ROO• react very easily with the double bond of unsaturated fatty acids to generate epoxides. In the second way, the molecule breaks and gives volatile and sensory active substances with less carbon atoms. Breaking the molecule takes place both due to the RO• created and depending on the position of the double bond in relation to the hydroperoxide group. From this, saturated and unsaturated aldehydes, saturated and unsaturated hydrocarbons, and oxo acids are formed. The aldehydes are most reactive compounds, which are further oxidized and react with the proteins. Malondialdehyde is an important product of this oxidation [87]. The third way is oxypolymerization, in which the number of carbons in the molecule is increased due to the reduction of two radicals. Concerning RO•, radicals are condensed by a -C-C- bond, which is not frequent, because RO• is less available.

The oxidation process catalyzed by metals, such as Fe and Cu, present in cellular components and tissues (e.g., muscle tissues), takes place by accepting an electron. They involve directly or indirectly in initiation, propagation, and termination reactions of radicals [88]. For example, exposure to heavy metals can change the composition of the reaction products. High concentrations of free radicals may outweigh termination reactions, where the metals inhibit the oxidation. Inhibition of oxidation may occur with higher concentrations of metal ions. It is believed that Fe and Cu ions oxidize and reduce hydrocarbon free radicals to their corresponding anions and cations together with the development of free radical complexes.

### 4.2. The Oxidation of Proteins

Proteins are a class of macromolecules that can perform a diverse range of functions for the cell. They help in metabolism by providing structural support and by acting as enzymes, carriers, or as hormones. The building blocks of proteins are amino acids. Proteins are organized at four levels: primary, secondary, tertiary, and quaternary. Protein shape and function are intricately linked and any change in shape caused by changes in temperature, pH, or chemical exposure may lead to protein denaturation and a loss of function.

### 4.3. The Oxidation of Carbohydrates

Living organisms are carbon-based because carbon plays a prominent role in the chemistry of living things. The four covalent bonding positions of the carbon atom can give rise to a wide diversity of compounds with many functions, accounting for the importance of carbon in living organisms. Carbohydrates are a group of macromolecules that are a vital energy source for the cell, provide structural support to many organisms, and can be found on the surface of the cell as receptors or for cell recognition. Carbohydrates are classified as monosaccharides, disaccharides, and polysaccharides, depending on the number of monomers in the molecule.

### 4.4. The Oxidation of DNA

Nucleic acids are molecules made up of repeating units of nucleotides that direct cellular activities such as cell division and protein synthesis. Each nucleotide is made up of a pentose sugar, a nitrogenous base, and a phosphate group. There are two types of nucleic acids, namely DNA and RNA.

DNA may be damaged, as a result of oxidative stress, from the reactions between hydroxyl radicals and all of the components of the DNA molecule, including the pyrimidine and purine bases and deoxyribose backbone. Studies have shown, in vitro, that ROS can further propagate DNA damage via the production of base-free sites, frame shifts, DNA-protein cross-linkages, deletions, and chromosomal rearrangements [71]. In addition, ROS may stimulate protein kinase and poly(ADP ribosylation) pathways, a process by which oxidative stress can impact on signal transduction pathways and modify the expression of genes [89]. These collectively may contribute to disease within the affected cells and tissues of a biological system. However, DNA damage, caused by free radicals, may be repaired by specific and non-specific repair mechanisms—albeit, there is the potential for the ill-repair of DNA, and this may contribute to disease, such as carcinogenesis.

## 5. Actions of Dietary Antioxidants, Endogenous Antioxidants, and Metal Binding Proteins on Free Radical Formation in Biological Systems

In biological systems, free radical formation is controlled by various defense systems, including enzymatic and non-enzymatic defense systems, metal binding proteins, and other endogenous antioxidants (Figure 8). Within any farm animal production system, overproduction of free radicals and oxidative stress is caused by overexposure of biomolecules to oxidants. The actions of defense systems on reducing free radical formation and alleviation of stress are important for better performance, health, and productivity in farm animals. Therefore, farm animals must be offered diets enriched in antioxidants, trace elements, and other bioactive substances to utilize major nutrients and energy in the diet in a synchronized manner.

### 5.1. Non-Enzyme Defence Systems

Non-enzymatic defense systems include scavenger antioxidants (vitamin C and E, glutathione, and coenzyme Q10) and some proteins which act as antioxidants by binding ROS and RNS. Examples of antioxidant proteins are SS-peptides, thioredoxin (Trx), and acute phase proteins, namely transferrin, albumin, ceruloplasmin, and haptoglobin. These antioxidant systems thus protect the tissues against ROS. Figure 9 demonstrate the antioxidant activities of vitamins, minerals, proteins, and phytonutrients.

#### 5.1.1. Antioxidant Actions of Vitamins

The most studied dietary antioxidants are vitamin E, vitamin C, vitamin B complex, and beta-carotene. Fruits and vegetables are significant sources of vitamin C and carotenoids, while whole grains and high quality green vegetative parts are major sources of vitamin E. Some natural sources of vitamin E for farm animals are fresh pasture, forage crops, legumes (e.g., lucerne), silage, yeasts, and bioactive plants and herbs. Vitamin E acts as an effective chain-breaking antioxidant within the cell membrane and prevents lipid peroxidation of membrane fatty acids as it is a major lipid-soluble antioxidant. Natural and synthetic forms of vitamin E can be supplemented to meat and milk producing animals, but natural tocopherol (d-α-tocopherol) is more effective in terms of transitioning from the diet into muscle than synthetic tocopheryl acetate, which is a mix of eight stereoisomeric forms [73]. Other antioxidants infer similar benefits to meat and milk quality, and while they can be provided within sheep, goats, or cattle diets, their contributions are relatively minor in comparison to that of vitamin E [74]. These include vitamin C, carotenoids, and phytonutrients (e.g., flavonoids, phenolic acids, tannins) compounds that are often available from the same dietary sources as listed for vitamin E.

Although, vitamin C may act as a prooxidant at high concentrations, it has the capacity to neutralize ROS in the aqueous phase before lipid peroxidation is initiated since vitamin C is the most important water-soluble antioxidant in extracellular fluids [75]. It has been stated that vitamin C is capable of regenerating vitamin E. Moreover, lipid-rich tissues are also protected by the antioxidant action of beta-carotene, lycopene, and other carotenoids, where beta-carotene may have synergistic activity together with vitamin E [82]. Absorption of beta carotene, vitamin E, and other fat-soluble nutrients is less with dietary sources having excessively low fat content.

Research indicates that B group vitamins have both antioxidant and prooxidant effects on lipid peroxidation, based on experimental conditions, exhibiting the activities in the early phase or later phase reactions. Some trials suggest that B group vitamins might have antioxidant effects on lipid peroxidation. For example, riboflavin (vitamin B2) may have an antioxidant action independently or as a component of the glutathione redox cycle. The antioxidant nature of riboflavin indicates that this vitamin can protect the body against oxidative stress, especially lipid peroxidation and reperfusion oxidative injury [77].

#### 5.1.2. Antioxidant Actions of Minerals and Trace Elements

Trace minerals are known as cellular prooxidant at higher physiological concentrations and are able to produce free radicals and cause toxicity. The metals are most effective in generating ROS upon reacting with oxygen. For example, free Fe^2+^ and Cu^2+^ ions can react with hydrogen peroxide and produce hydroxyl radicals via the Fenton reaction; a reaction between free iron and H_2_O_2_ that forms hydroxyl and hydroxyl radical [83]. Similarly, copper can interact with H_2_O_2_ to generate hydroxyl radicals 60 times faster than that of iron. On the other hand, trace minerals are incorporated in metal-binding proteins, namely transferrin, haptoglobin, lactoferrin, metallothionein, ferritin, hemopexin, ceruloplasmin, myoglobin, and albumin, which are a vital part of the extracellular antioxidant defense system. These minerals play an essential role as cofactors of necessary antioxidant enzymes. For example, selenium (Se), mainly through its incorporation into selenoprotein, is a cofactor for selenoenzymes [90] and plays a crucial role in regenerating other antioxidants, including vitamin C.

#### 5.1.3. Antioxidant Actions of Proteins

Many proteins act as a part of the non-enzymatic antioxidant system of the body. Proteins such as ceruloplasmin, lactoferrin, albumin, haptoglobin and hemopexin, SS peptides, and thioredoxin are considered as antioxidants. Ceruloplasmin, an abundant Cu-containing enzyme, facilities iron loading onto transferrin by catalyzing the oxidation of ferrous (Fe^2+^) ions to ferric (Fe^3+^) ions in the serum. Furthermore, ceruloplasmin prevents lipid oxidation, scavenges superoxide, and binds free copper ions. Albumin, a highly soluble protein, represents the most abundant and the predominant antioxidant agent in the plasma. Albumin sequesters copper and iron in plasma and makes them less susceptible to participate in the Fenton reaction. Albumin scavenges some hydroxyl radicals produced from both copper and iron reactions with H_2_O_2_. Serum albumin is responsible for over 70% of the free radical-trapping activity of serum [91]. Lactoferrin is an iron-binding glycosylated protein found in mammalian milk, tears, and saliva. It is structurally similar to transferrin and can chelate two ferric irons (Fe^3+^), but unlike transferrin, lactoferrin does not release its iron even at low pH. This property enables lactoferrin to limit the availability of iron to microbes in infected tissues. Iron generated by the liver or macrophages leaves the cells or tissue as ferrous iron (Fe^2+^) and plasma ferroxidase ceruloplasmin converts the Fe^2+^ to ferric iron (Fe^3+^). Ferric iron produced by the ferroxidase activity of ceruloplasmin could be taken up by either transferrin or lactoferrins. A direct transfer of ferric iron between the ceruloplasmin and lactoferrin could prevent both the utilization of iron by pathogenic bacteria and the generation of toxic hydroxyl radicals [92]. Metallothionein are a family of small, cysteine-rich metal-binding proteins that are important for zinc and copper homeostasis [93]. Metallothionein have high affinity for Cu^+^, thus avoid formation of ROS by stopping the reaction of free Fe^2+^ and Cu^2+^ ions with hydrogen peroxide in the animal or human body. Metallothionein not only stores metal ions but also functions as a metal buffer for zinc and copper metabolism under physiological conditions [94]. Haptoglobin and hemopexin are iron-binding proteins that are synthesized in the liver and released into the circulation. The major role of haptoglobin is to bind free hemoglobin and heme released from lysed-erythrocytes and to transport it to the liver for catabolism and excretion. These proteins prevent the reaction of iron released from hemoglobin with molecular oxygen [95]. Thioredoxin, a small thiol-active protein, is one of the central antioxidant systems in mammalian cells. It maintains a reducing environment by catalyzing electron flux from NADPH through thioredoxin reductase to thioredoxin, which reduces its target proteins using highly conserved thiol groups [86]. Thioredoxin is effective at trapping hydrogen peroxide and hydroxyl radicals. The SS (Szeto–Schiller) peptides are important in antioxidant supply to the inner mitochondrial membrane. These SS peptides are capable of scavenging hydrogen peroxide and peroxynitrite and inhibiting lipid peroxidation [96].

#### 5.1.4. Antioxidant Actions of Phytonutrients

There are several other dietary antioxidant substances that exist besides the vitamins, minerals, and proteins discussed above. Numerous plant-based components, collectively known as ‘phytonutrients’, or ‘phytochemicals’, are gaining attention owning to their antioxidant activity [97,98]. Polyphenol compounds, such as flavonoids or phenolic acids, are abundant within the plant kingdom and >1000 substances have been identified [99]. In plants, flavonoids and other phytonutrients act as a defense system against a broad array of environmental stress factors. Whereas, in the biological systems of animals and human, phytonutrients (e.g., flavonoids) serve as modifiers or defense mediators with potential anti-inflammatory, anti-carcinogenic, antiallergenic, anti-viral, and anti-ageing activities [100]. As an example, antioxidant properties of flavonoids are largely responsible for their promising therapeutic effects in human [98]. Besides, flavonoids may exhibit protective effect towards heart disease via inhibiting cyclooxygenase and lipoxygenase activities of platelets and macrophages [100,101].

In animal nutrition, particularly for monogastric species (poultry and swine), polyphenol-rich plant extracts have been broadly explored as potential sources of natural antioxidants to substitute their synthetic counterparts [102]. The utilization of by-products as feed supplements from the orchard (wine) industry or oil (olive) industry or citrus industry is of interest for many researchers since their use in animal nutrition may enhance their health, productivity, and meat quality [103], meanwhile providing cheap feed supplements or by-products from human food produce. At the same time, the inclusion of these by-products as part of animal feeds may have a positive impact on the environment by lowering the risk of phytotoxic phenomena from their deposition on soil and natural water reserves [104]. For example, the inclusion of grape by-products, which are rich in polyphenols, has enhanced the oxidative stability of the meat and the overall nutritional and sensory quality of meat and meat products, and thus extended their shelf-life [103]. It is important to highlight that flavonoid compounds are poorly absorbed in the gut and their concentrations in target tissues are often too low to perform an effective antioxidant defense as reviewed by Surai [105]. For humans, consumption of various plant foods provides many phytochemicals that may significantly impact intestinal health, however, at present, no dietary recommendation for humans for phytochemicals or polyphenols exists [106,107].

Recently, there has been a large number of research articles or review reports published about the availability of antioxidant contents as phytonutrients in fruits (sap and peels), vegetables, grains/seeds, nuts, and yams [108,109,110]. Most of these papers report the total antioxidant levels, their activities, and the methods for quantification. It is true that these phytonutrients, for example, flavonoids or lignans, may play some roles in improving the blood lipid profiles in terms of cardiovascular health, improving the pre- or post-menopausal hormonal status in women health, or inhibiting the signal transduction and tissue proliferation in early stage of cancer development in humans. These outcomes were mainly found in blood related assays or findings. The latter activities were categorized as antioxidant defense, alleviation of oxidative stress, and health enhancement in humans.

The reality is that the actions and functions of these phytonutrients, such as flavonoids, tannins, or phenolic acids, and phytic acid, have not been fully investigated in terms of their presence in tissue systems of muscles and milking farm animals that is attributable to the enhancement in quality and/or preservative aspects of meat and milk. There were not many convincing well-designed in vivo animal experimental studies conducted in farm animals (sheep, goat, or dairy and beef cattle) with presentation of validated data of the existence (availability) of these phytonutrients in animal products, such as meat or milk. Future research with statistically designed animal studies is vital indeed to prove that whether feeding these phytonutrients (or consumption of feeds containing these phytonutrients by farm animals) to animals can increase the deposition of these compounds (e.g., flavonoids) in muscle tissues of farm animals and their relationship with enhanced meat or milk quality. The effect is to be shown as quantification of these phytonutrients in the tissue systems and their resultant effect on improved meat and milk aspects provided with regression or correlation analytical matrix outputs [15,111]. At the current stage, the mechanisms behind the effects of phytonutrients on antioxidant action of muscle tissues in farm animals and improvement in meat and milk quality can be classified as speculation only, we believe, and this is attributed to the fact that polyphenols have a low bioavailability. In this context, it should be noted that the status of vitamin E [112,113,114] or vitamin C and Se [115] or Zn [116] as minerals on antioxidant action of muscle tissues in farm animals and improvement in meat and milk quality have been very well documented with well-designed in vivo experiments conducted in farm animals [32]. For example, supplementation of vitamin A or C to lactating ewes from late pregnancy to mid-lactation enhanced milk quality by increasing the levels of healthy FAs and the antioxidant capacity (in terms of increased DPPH scavenging activity and reduced MDA concentration) of the milk [117]. Intramuscular vitamin E injection has decreased lipid oxidation, preserved the redness of meat, and increased the nutritional value and consumer acceptability of lamb meat [118].

A recent study indicated that feeding diets rich in phytonutrients (terpenoids and alkaloids) have beneficial effects on improving the performance of livestock, believed to be through improved antioxidant action [119]. Feeding a distilled rosemary by-product to lamb has enhanced the contents of rosmarinic acid, carnosol, and carnosic acid in the meat. In addition, fresh meat had lower DPPH values and higher total ferric reducing antioxidant power compared to controls, indicating susceptibility to oxidation in lamb meat [120]. Similarly, meat from lambs fed oregano essential oil (1 mL/kg DM) showed less susceptibility to lipid oxidation during both refrigerated and frozen storage compared to control groups [121]. Gobert et al. [122] showed that the combination of vitamin E and polyphenol rich plant extracts was more capable of enhancing lipid stability of beef steaks than vitamin E alone, indicating a possible synergetic effect. Dietary supplementation with plant extracts has been associated with lower MDA levels [123].

Carotenoids are other phytochemicals may be categorized into two major groups: (1) xanthophylls, oxygenated carotenoids (lutein, zeaxanthin, and b-cryptoxanthin); and (2) carotenes, hydrocarbon carotenoids that are either cyclized (α-carotene and β -carotene), or linear (lycopene). In forages, nearly 10 carotenoid compounds (i.e., xanthophylls and carotene) have been identified [124]. Antioxidant activity of lycopene, lutein, canthaxanthin, and zeaxanthin are similar to, or even greater than, those of beta-carotene, even though they do not possess pro-vitamin A activity. It is believed that carotenoids (i.e., xanthophylls and carotene) present in forages may contribute to the nutritional and sensory characteristics of milk and meat products. Carotenoids are potent antioxidants of plant origins. For example, lycopene, which is found in red-color fruits (such as tomato), has shown robust antioxidant activity [125]. Lycopene has been reported as the most efficient antioxidant among carotenoids [126]. Dietary lycopene supplementation has reduced thiobarbituric acid reactive substances (TBARS as units of malondialdehyde (MDA), reflecting a lower degree of lipid and protein oxidation [127].

Phytic acid is another phytochemical, constituting 1–5% of most cereals, nuts, legumes, oil seeds, pollen, and spores. Phytic acid may serve as a promising antioxidant because it exhibits a high chelation potential and can be supplemented to diets to inhibit iron-catalyzed oxidative reactions [128].

### 5.2. Antioxidant Defence by Metal Binding Proteins

#### 5.2.1. Actions of Iron in Biological Systems

In biological systems, iron is found as ferrous (Fe^+2^), ferric (Fe^+3^), and ferryl (Fe^+4^) states. Ferrous (Fe^+2^) is the form that participates in biochemical reactions. In anoxic environments, ferrous (Fe^2+^) is reasonably soluble at neutral pH, but in the presence of oxygen, aqueous Fe^2+^ is rapidly converted to the relatively insoluble ferric (Fe^3+^). Ferric iron can be solubilized at pH < 3. Animals acquire ferrous iron by acidifying the environment to facilitate the solubilization of Fe^3+^, followed by the reduction of Fe^3+^ to Fe^2+^. Exposure to low pH of the stomach helps maintain iron in the solubilization of Fe^3+^ and iron absorption occurs in the upper portion of the intestine (duodenum). Then, intestinal cell surface reductases reduce Fe^3+^ to Fe^2+^. Ferrous iron is exported from the intestine by a protein transporter: ferroportin. Ferroportin-exported iron is rapidly converted to Fe^3+^ bound to ferritin. In the blood, ferric iron (Fe^3+^) is transported by transferrin. Iron-binding transferrin is highly selective for Fe^3+^ and can bind two atoms of ferric iron. When erythrocytes lyses and hemoglobin or heme is released, hemoglobin is captured by haptoglobin and heme by hemopexin. Cellular uptake of iron and dissociation of iron from transferrin occurs in combination of low pH, binding of transferrin to its receptor and reduction of Fe^3+^ to Fe^2+^. When the binding capacity of transferrin is exceeded by iron entering the blood, iron is cleared by parenchymal cells, such as hepatocytes, in the liver. Animals store iron in the form of unreactive ferric ion (Fe^3+^) bound to ferritin, which tightly binds Fe^3+^. Soluble iron enzymatically is converted to ferric ion (Fe^3+^) in ferritin. Iron stored in ferritin as iron bound to transferrin is inert and does not generate ROS. Coordination of iron entry into blood with iron utilization and storage is regulated by small peptides termed hepcidin. Hepatocytes and macrophages synthesize and release the hormone hepcidin which acts in the small intestine to decrease iron absorption. In general, hepcidin increases by iron overload and decreases by iron demand [129]. Feeding diets rich in iron is important for active muscle contraction and energy metabolism in the animal as iron in the form of hemoglobin carries oxygen to the cellular- and muscle-systems. Sow milk contains low iron concentrations and suckling pigs therefore receive supplemental iron, either via intramuscular injection or oral supplementation. A recent study showed that iron deficiency in pigs led to anemia and lower ceruloplasmin activities, however, biochemical indices of oxidative status were comparable with the iron supplemented pigs [130]. Iron excess is believed to generate oxidative stress, however, the amount of iron within the cell is carefully regulated in order to provide an adequate level of this micronutrient while preventing its accumulation to the toxic level. Iron toxicosis is relatively uncommon in pigs but has been reported sporadically after iron injections of piglets, and incidents have sometimes been associated with vitamin E or selenium deficiency in the sows [131]. Upon slaughter of animals, when the muscle is converted to meat, iron in the state of hemoglobin is converted to the myoglobin state, which is an important component to keep meat in cherry red or bright red. Thus, the incorporation of antioxidants into muscle meat can be important to stabilize meat color, especially in red meat animal species.

#### 5.2.2. Actions of Copper in Biological Systems

In biological systems, copper (Cu) ions can exist in reduced (Cu^2+^) and oxidized (Cu^1+^) states. Reduced copper prefers oxygen donors (such as glutamate or aspartate) and nitrogen donors (such as cysteine), while oxidized copper prefers sulfur donor ligands, such as cysteine or methionine [132]. Cu^+1^ is unstable in aqueous solution and has to be either stabilized by binding to carriers or oxidized to a more stable form (Cu^2+^). Dietary Cu^2+^ is reduced to Cu^1+^ by a metalloreductase in the lumen of the gut and then transported into intestinal cells through a Cu^1+^ transporter (Ctr1). The majority (70%) of Cu intake of mammalian cells is mediated by Ctr1 [133]. Cu^1+^ from intestinal cells pumped out via Cu-transporting ATPase, termed ATP7A, which is expressed in several tissues. In the bloodstream, Cu^1+^ is transported via the portal vein to the liver, the central organ for Cu hemostasis. In the liver, Cu is incorporated into ceruloplasmin and then secreted in plasma. Excess Cu is exported out by another Cu-transporting ATPase (ATP7B), which is expressed in the liver. In serum, most of the Cu is bound to ceruloplasmin and the rest is bound to albumin, transcuprein (β-macroglobulin), and low molecules carriers [133]. Previously, there was a general consensus that ceruplasmin is the potential Cu-carrier, however, results from rodent studies argue against the role of ceruloplasmin in Cu delivery to tissues [134]. CuSO_4_ has been used to enhance performance in pigs, and that it could (partly) be due to antioxidant actions of Cu via the expression of SOD activity in the tissue systems. Recently, it has been suggested that small Cu carriers, SCC, play a significant role in regulation of Cu delivery to tissues [135], however, it should be noted that the cellular and systemic components that function in Cu homeostasis are not fully understood [135].

### 5.3. Enzymatic Defence Systems

Enzymatic defense systems, namely superoxide dismutase (SOD), catalases (CAT), glutathione peroxidases (GPx), glutation reductases, and glutathione transferases, protect DNA and mitochondria from oxidative stress damages. It has been stated that polymorphisms in these enzymes are related to the DNA damage and consequently affects an individual’s defense against the development of numerous diseases [40,136]. Antioxidant enzymes can stabilize or deactivate free radicals before they are reacting with biomolecules or attacking cellular components. They reduce the energy of the free radicals or provide some of their electrons to free radicals, thereby making them (free radicals) stable. Furthermore, they minimize the damage caused by the free radicals via interrupting the aspects of antioxidant enzymes. Free radicals are significantly oxidizing a chain reaction. Over the past decade, many studies have revealed the positive association of more than sixty different health conditions in humans, including ageing process, diabetes, cancer, strokes, Alzheimer’s disease, heart attacks, and atherosclerosis.

#### 5.3.1. Superoxide Dismutase and Superoxide Reductase

Superoxide dismutase (SODs) and superoxide reductase (SORs) are metalloenzymes that catalyze the dismutation of superoxide radicals to oxygen (O_2_) and hydrogen peroxide (H_2_O_2_). SODs have been found in all types of living organisms. By contrast, ROS have been found only in prokaryotes and recently, in unicellular eukaryotes. SODs contain Mn, Fe, Cu, Zn, and Ni, while SORs only contain iron. SODs require one proton per superoxide and no external reductant while SORs need two protons per superoxide as well as an external reductant to provide the electron [137]. SODs exist in different isoforms: systolic CuZnSOD, mitochondrial MnSOD, and extracellular SOD (EC SOD). The extracellular SOD is a tetrameric, high molecular weight enzyme, found in several extracellular fluids, including plasma, lymph, and synovial fluid. The overall reaction catalyzed by SODs is given below:2O_2_^−●^ + 2H^+^ → H_2_O_2_ + O_2_

#### 5.3.2. Glutathione Peroxidase

The glutathione peroxidases (GPx) are expressed in eight isoforms (GPx1–8) and use glutathione (GHS) to catalyze the reduction of H_2_O_2_ to water. GPx containing selenium (GPx1–4 and 6) or not-containing Se (GPx5, 7, and 8) is present at different tissues and cellular compartments. GPx1 and GPx4 are expressed in most tissues. GPx2 is expressed in the gastrointestinal tract and GPx3 is synthesized in kidney and secreted for use in plasma. The GPx enzymes utilize selenium to detoxify ROS. GPx plays an important function in the metabolism of tripeptide glutathione [102], which is one of the most significant intracellular antioxidative defense mechanisms. Once peroxides are converted to water (or alcohol), GSH is simultaneously oxidized to GSSG (oxidized form of glutathione). The reactions of GPx and GSH with both peroxides are given below. It is important to note that this reaction is expensive, that is, removal of one H_2_O_2_ molecule requires two molecules of the valuable GSH. First, a hydroperoxide (ROOH) oxidizes selenol in GPx to selenenic acid. Then, the first GSH forms a selenadisulfide with the selenenic acid, and the oxygen is removed as H_2_O. Finally, the second GSH reduces the selenadisulfide by a thiol–disulphide exchange. Thereby, GSSG is released and the enzyme is regenerated to the selenol form, which is now ready for the next cycle [138].
GPx-Se^−^ +H_2_O_2_ → GPx-SeOH + OH^−^
GPx-SeO^−^ + H^+^ + GHS → GPx-Se-SG + H_2_O
GPx-Se-SG^−^ + GHS → GPx-Se^−^ + H^+^ + GSSG

#### 5.3.3. Catalase

Catalases are efficient intracellular antioxidant enzymes that catalyze the conversion of two hydrogen peroxide molecules to water and oxygen. One molecule catalase is capable of catalyzing six million molecules of H_2_O_2_ each minute.

#### 5.3.4. Heme Oxygenase

Heme oxygenase, existing in two isoforms, catalyzes the degradation of heme and produces biliverdin, ferrous iron, and carbon monoxide. Afterward, the biliverdin is converted to bilirubin by biliverdin reductase enzyme. Bilirubin acts as an antioxidant, when it is oxidized to biliverdin and then recycled by biliverdin reductase back to bilirubin [139].

### 5.4. Other Endogenous Antioxidants

#### 5.4.1. Glutathione

Glutathione is an intracellular low molecular weight tripeptide (g-glutamyl-cysteinyl-glycine) found in a high concentration in cytosol of all mammalian cells. Glutathione exists in both reduced and oxidized form (GSSG) [102]. Glutathione, similar to thioredoxin and metallothionein, chelates metals such as zinc, copper, mercury, and cadmium via its cysteine residue. GSH is a major part of the cellular defense system against exogenous and/or endogenous oxidants. Free radicals and ROSs (hydroxyl radical, lipid peroxyl radicals, and peroxynitrite) easily convert GSH in GSSG, though the GSH:GSSG ratio is always maintained at ≥10. GSH provides the reducing power for the reaction of GPx when catalyzing the breakdown of H_2_O_2_ (two GHS is utilized). When GHS is oxidized by GPx or ROSs, it is reduced back by glutathione reductase consuming NADPH [140]. Dietary glutathione sources include green foods, asparagus, avocado, cucumber, green beans, and spinach [141].

#### 5.4.2. Lipoic Acid

Lipoic acid (1,2-dithiolane-3-pentanoic acid), also called thioctic acid, acts as antioxidant. It is synthesized in the mitochondria from octanoic acid (C8:0) and cysteine. Lipoic acid contains two vicinal sulfur atoms that can undergo redox reactions; the reduced form of lipoic acid (dihydrolipoic acid) has two thiols. Lipoic acid functions as an amphipathic antioxidant with the capacity to quench free radicals and regenerate other cellular antioxidants. Lipoic acid involves in cellular antioxidant protection through both direct and indirect functions. Dihydrolipoic acid is able to reduce oxidized forms of other cellular antioxidants, including glutathione, α-tocopherol, ascorbic acid, as well as quenching ROS and RNS. Lipoic acid can chelate free iron and copper, preventing them from generating ROS. Lipoic acid in food binds to lysine (lipoyllysine) and the main sources of lipoic acid are animal meat (kidney, heart, and liver) and vegetables such as spinach and broccoli [142]. Several studies have reported that dietary supplementation of lipoic acid enhances antioxidant capability and storability of ruminant and poultry meat and meat products [143,144].

#### 5.4.3. Uric Acid

Uric acid is produced by a xanthine oxidoreductase as a by-product of purine metabolism, which has both antioxidant and prooxidant properties. Uric acid is an abundant aqueous antioxidant that comprises almost two-thirds of all free radical scavenging activity in human serum. A potential role of uric acid as an antioxidant has been speculated in relation to its interaction with vitamin C and iron.

#### 5.4.4. Coenzyme Q_10_

Coenzyme Q_10_ or ubiquinone is a vitamin-like substance in the electron transport chain of the mitochondrial membrane, and its reduced form acts as a part of the intracellular antioxidant system to protect phospholipids and membrane proteins against free radicals [145]. Primary dietary sources of Coenzyme Q_10_ are oily fish, organ meats such as liver, but lower levels can be found in most dairy products, vegetables, fruits, and whole grains [146]. The reduced form of coenzyme Q_10_ (ubiquinol) is known as a lipid-soluble potent antioxidant that has roles in donating electrons to the ROSs in regenerating other antioxidants, such as tocopherol and vitamin C, to their active form [147].

## 6. Feeding Systems, Antioxidant Actions and Their Impact on Productivity and Functional Aspects of Meat and Milk in Farm Animals

The world has seen advanced research in the application of dietary antioxidants and their biological actions to remediate free radical damage, oxidative stress, and human health. This has given hope to farmers, producers, and animal researchers who aim to improve animal performance, productivity, and product quality in livestock by supplementing diets. Alternatively, dietary antioxidants can be also offered through the adoption of appropriate feeding systems that alleviate oxidative stress, subclinical disease development, and poor immune function. There is a need for nutritional interventions to limit production of ROS, oxidative stress, and inflammatory actions so that the productivity and health of farm animals can be optimized. Some production systems offer different levels of nutrients to the farm animals that may protect from or cause oxidative stress (Table 2).

There is a lack of knowledge regarding the burden of oxidative stress on the progression of infectious diseases in the gastrointestinal tract of pigs and poultry [5,45], and other infectious diseases that develop in goats, sheep, and cattle during the pregnancy–lactation stage. The latter is obvious when nutritional deficiencies are encountered as a result of climate variability, such as drought or the overload of parasites infestation. Understanding the means of oxidative stress reactions in farm animals may help to alleviate the impact on animal health [148] and ultimately, the quality of their products, such as meat and milk, for human consumption. In the following sections, the role of dietary antioxidants in the form of vitamins, minerals, fatty acids, and other bio-actives (phytonutrients) will be discussed. This is in accordance with their effect on feed intake, animal health, or impaired immune function that may infer productivity losses and quality deterioration of meat and milk products. It is well known that on-farm and off-farm factors such as animal species, genetics, diet, gender, age, storage length, packing type, and cooking methods can modify the product quality of meat and milk. Among these, the nutritional background of an animal (feeding systems) has the greatest influence on the oxidative stability and preservative (functional) properties of milk and meat [149]. Figure 10 illustrates the impact of dietary antioxidants or antioxidants supplementation on the overall performance and meat and milk production in farm animals.

### 6.1. The Impact of Dietary Antioxidants on Performance and Productivity of Farm Animals

When the metabolic rate of the individual is high (animals with faster growth rate, greater milk production, etc.), it is more prone to oxidative stress due to the generation of oxygen radicals. Reproductive mechanisms may also be linked to oxidative stress, i.e., as described for sow reproduction, and gestation can be considered to be a state of oxidative stress arising from increased placental mitochondrial activity and production of ROS [105]. Thus, under common live responses (growth, reproduction), supplementation with exogenous antioxidants is required to provide the redox homeostasis in cells and tissues of biological systems, as shown in Figure 3 and Figure 4. Livestock may access dietary antioxidants from the origin of flora (forages, crops, and grains) since several plant products are rich in delivering exogenous or endogenous antioxidants generated from macro-micro-nutrients or phytochemicals. Therefore, the provision of antioxidants, as diets or supplements for farm animals, can be a valuable practice to maintain animal health and optimize flock or herd productivity [150,151].

Young animals, such as piglets post weaning and broiler chickens, have immature intestines and immune functions which make them vulnerable to invading microorganisms and disease. Various intestinal diseases can cause damage to the intestinal epithelium and consequently, restrict the absorption of nutrients. Some examples of these diseases in poultry include necrotic enteritis, malabsorption syndromes, and coccidiosis [152,153]. In pigs, when suckling milk from the sow, the fat digestion is high at ~96% but this rate is reduced during the postweaning period to ~65% [154]. It was stated that weaned piglets, challenged with *E. coli* bacteria, had impaired vitamin E status and this is probably caused by a combination of reduced fat digestion and increased α-tocopherol demand, as an antioxidant for immune function [5]. Provision of organic selenium in sow and piglet nutrition has also been shown to improve antioxidant status of the animals [105].

Infectious diseases, such as pneumonia and enteritis, in farm animals are thought to be associated with oxidative stress. Esmaeilnejad et al. [155] evaluated the antioxidant status and oxidative stress in sheep naturally infected with *Babesia ovis*. Red blood cell (RBC) count, hemoglobin (Hb) concentration, packed cell volume (PCV), piroplasm parasitemia percentage, malondialdehyde (MDA) concentration, erythrocyte superoxide dismutase (SOD), glutathione peroxidase (GSH-Px), catalase (CAT) activities, and total antioxidant capacity were determined in sheep naturally infected with *B. ovis* and healthy non-infected animals. Microscopic examination of Giemsa-stained peripheral blood smears revealed *B. ovis* infection. Compared to controls, the activities of erythrocyte GSH-Px, SOD, TAC, and CAT showed a significant decrease, whereas the concentration of MDA in erythrocytes of infected sheep increased significantly. The parasitemia rate was positively correlated with MDA and negatively correlated with PCV, SOD, CAT, GSH-Px, and TAC. It was concluded that *B. ovis* plays an important role in the occurrence of oxidative damage to RBCs and anemia in ovine babesiosis [155].

The nutrient imbalances and form of lipid peroxides from dietary fat oxidation can also trigger oxidative stress and inflammatory reactions and therefore can affect gut health, immunity, growth, and the development of livestock [11]. Ruminants (cattle, sheep and goats, alpaca, rabbit, horse) have complex stomach systems that facilitate their consumption of a fibrous based forage diet. Due to the involvement of microbiota in fiber degradation, ruminant animals are accustomed to ingesting diets containing < 4% of lipids on a dry matter basis [156]. Anything > 6% lipid in the diet can be toxic to rumen microbes and interfere with microbial activity, such as decreased rates of lipolysis and biohydrogenation in the rumen that can lead to lower feed intakes and nutrient digestibility [157,158,159]. On the other hand, when animals are fed with a diet rich in fat and oil, the free radicals resulting from the products of peroxidation in the circulatory systems will attack the tissue systems and can generate oxidative stress. This can cause inefficient energy metabolism and nutrient partition in the body that in turn contributes to lower meat or milk production. In this situation, the provision of a diet rich in antioxidants is vital for lowering lipid oxidation in the body, allowing consistent productivity of high performing farm animals.

Radiation on the other hand may induce free radical formation in two ways: (1) through the degradation of nutrients, such as essential lipids, vitamins, and other bioactive compounds from forages, such as grasses and legumes, due to exposure to extreme temperatures and dry conditions associated with climate variation; and (2) direct solar radiation and high heat conditions causing animals oxidative stress and ill health. Ultraviolet rays can degrade the skin and mucous membranes of unsheltered animals. Excess infrared rays during hot and sunny periods also cause oxidative stress as a result of heat stress. This can affect the animal’s basal metabolism and growth performance mostly due to low feed intake and dehydration [28]. Previous research indicates that skin lesions and heat stroke are common incidents in goats, sheep with short wool, and European cattle with less hair due to high temperature and high radiation [160]. Such conditions can cause reduced feed intake and nutrient partition in the body, resulting in lower weight gain and productivity in farm animals grown under grazing conditions.

During summer-autumn seasons, pastures dry off, decay, and lose their nutritional value due to high heat, solar radiation, and low moisture. As a result, forage will offer lower vitamins, essential fatty acids, and proteins and would not satisfy the nutrient requirement of growing lambs or beef cattle. Under these circumstances, supplementary feeding with other cereal grains or protein meals is necessary to boost the productivity of farm animals. Sheep grazing senesced perennial pastures containing mainly lucerne during autumn had greater liveweight gain and carcass weight than those counterparts grazing senesced annal ryegrass pastures with supplements of cereal grain or cracked flaxseed [161]. As spring progresses, the pastures species matures at different rates and the crude protein and energy concentrations in the feed decline. Such conditions reduced the liveweights and carcass weights of lambs grazing ryegrass pasture but not lucerne pasture at slaughter [162]. A recent study showed that the growth rates and carcass traits of sheep supplemented with lucerne hay at 55% in a total mixed ration were similar to that achieved with a diet formulated with cereal grains at 55%, although the lucerne diet was lower in ME concentration [163]. These superior growth rates and productivity outcomes in sheep fed lucerne pasture or lucerne hay were believed to be the fate of vitamins, proteins, and essential fatty acids found in lucerne forage when encountering microbial action within the rumen and subsequent digestion. Specifically, the essential fatty acids and fat-soluble vitamins present in lucerne may bypass the rumen and enter into the circulatory systems, thereby improving redox status and dietary nutrient use in the body. This phenomenon helps to improve the availability of dietary energy and other nutrients needed to support greater productivity. Human studies have shown promising outcomes whereby dietary essential fatty acids (ALA, EPA, and DHA) and vitamins (E and C) can elevate health status by improving immune status, inflammation, and metabolic disorders in the body [164,165,166] and these responses are associated with improved antioxidant defense and nutrient utilization in the body.

There is evidence in sheep, poultry, and swine that omega-3 fatty acids offer antioxidant properties that improve the immune status and have proinflammatory effects in animals [5,159,167]. Most research on oils and fat as animal feeds, in monogastric (poultry and swine) and ruminant (sheep, goats, and cattle) nutrition, has been attributed to the understanding of the importance of lipids as a dietary energy source. However, some fatty acids, mainly PUFA, are known to be essential fatty acids and may act as bioactive nutrients influencing many important pathophysiological processes—for example, lipid metabolism, second messenger signal transductions and gene expression, and immune function and disease prevention. It is recognized that animals require the essential fatty acids linoleic acid (C18:2n-6) and α-linolenic acid (C18:3n-3), which are the respective building blocks of the n-6 and n-3 series, and precursors for the eicosanoids production in the biological systems. By desaturation and elongation processes, linoleic acid (C18:2n-6) forms arachidonic acid (C20:4n-6). By the action of cyclooxygenase, C20:4n-6 is converted to prostaglandins, whereas the action of lipoxygenase converts the n-6 FA into leukotrienes and other oxidative products [159,168]. These eicosanoid products of the n-6 series are believed to be proinflammatory and have been found to exert clinical efficacy in human diseases, including inflammation. On the other hand, linolenic acid is formed to eicosapentaenoic acid (EPA C20:5n-3), which can be further metabolized through desaturation and elongation into docosahexaenoic acid (DHA C22:6n-3). These eicosanoids act as anti-inflammatory and have a molecular structure similar to the ones originating from arachidonic acid (ARA), but their biological activities are different in terms of generating oxidative stress and inflammatory aspects in the body.

### 6.2. The Impact of Dietary Antioxidants on Functional Aspects of Meat from Farm Animals

In the last two decades, there has been a growing interest in feeding animals with diets rich in antioxidant compounds to achieve potential health benefits to both animal and those who consume animal products [151]. The inclusion of by-products containing antioxidants and other bioactive compounds from agricultural industries in the diets of farm animals has been on the rise because: (1) they contain valuable nutrients as vitamins, minerals, and essential fatty acids, which can enhance animal performance and product quality; (2) they are available for use as animal feeds at lower prices compared to traditionally used cereal grains or protein meals; (3) availability of improved technologies for processing and preservation of animal feeds with by-products to eliminate microbial growth related health issues in animals; and (4) protection of the ecosystem when these residues are discarded in terrestrial and aquatic ecosystems, potentially causing environmental problems and economical losses. However, there are limitations to the inclusion of by-products into animal feeds: (1) as these can increase heavy metals or chemical residues that can translate to the animal body, which may cause health retardation or product safety; and (2) they can easily be a growing media for fungus and mold, which can retard animal growth or even be toxic to animal health at a high rate of feed consumption. Therefore, caution must be taken with the purity of the by-products and level of inclusion in the animal diet as mixed rations. In this context, the use of grape pomace, olive cake, and distillers grain residues has been used in animal industries around the world, mainly in Europe and south or central American countries due to their availability and further processing capabilities.

Recent research studies performed in animals in vivo have reported that by-products rich in polyphenols, generated from olive oil and the grape wine industry, improved antioxidant capacity, meat quality, or welfare of productive animals, such as chickens, pigs, and lambs [21,169,170]. One study indicated that the administration of feed supplemented with grape pomace to weaned piglets enhanced antioxidant mechanisms, prevented oxidative stress damage to lipids and proteins, and improved the gut barrier function and health, thus the authors concluded this feed type had beneficial effects for animal welfare [171]. The piglets supplemented with grape pomace, had significantly increased antioxidant mechanisms in almost all of their tissues (e.g., heart, kidney, liver, quadriceps muscle, and stomach) as shown by comparative increases in GSH, H_2_O_2_ decomposition activity, and total antioxidant capacity. In addition, grape pomace inclusion in the diet of piglets improved their performance, by increasing body weight and enriching meat with n-3 PUFA. Supplementation of condense tannins has inhibited lipid peroxidation and improved the antioxidative status of transition dairy cows [172]. Dietary pomegranate by-products supplementation, in particular PSP, could improve antioxidant status, which was associated with a decline in lipid oxidation (FFA and β-hydroxybutyrate) and peroxidation (MDA) in dairy cows [173]. Polyphenols have been reported (1) to interrupt the propagation stage of lipid autoxidation chain reactions as effective radical scavengers; and (2) to act as metal chelators to convert hydroperoxides or metal prooxidants into stable compounds with the consequent decrease in reactive ^•^OH caused by the Fenton reaction [174].

There is evidence that increased absorption of polyphenol with supplementation can exert a protective effect on α-tocopherols by acting as a defensive barrier against lipid oxidation from oxidative decay [175]. Tufarelli et al. [176] found an improved antioxidant defense system and a reduced TBARS level in chicken liver following dietary supplementation with extra virgin olive oil. Branciari et al. [169] reported the beneficial effects of olive polyphenols on the oxidative status of meat measured through TBARS, demonstrating their antioxidant effect in meat of animals fed olive phenolic compounds. It could be argued that vegetable oils are mostly rich in vitamin E and the combined effect of polyphenols and vitamin E might be the reason for lower TBARS in the tissues, a protection of lipid oxidation. Others have demonstrated that olive mill wastewater, an olive polyphenol, when used in chicken diets can delay meat lipid and protein oxidation without affecting its color stability [104]. It was stated that the effect of dietary olive mill wastewater on meat antioxidant activity might be due the presence of bioactive molecules in muscle and liver tissues of animals fed with olive oil polyphenols [169].

Research by Kafantaris et al. [21] showed that the inclusion of grape pomace in the diet of growing lambs improved growth performance by increasing body weight as well as enriching lamb meat with n-3 PUFA. There was a significant increase on the content of EPA and DHA while lowering the n-6/n-3 ratio in meat compared to the control group. Inclusion of grape pomace in the diets of lambs also affected protein oxidation and lipid peroxidation in the quadriceps muscle at 42 and 70 days of ageing, observed as a slight decrease in TBARS and protein carbonyls when compared with the control group [21].

Feeding native or novel forages to ruminants as pastures, fodders, silage, or haylage has shown promising outcomes in improving the quality and preservative aspects of meat from goat, sheep, and cattle, mainly through the antioxidant activities in animal tissues driven by vitamins, minerals, and/or essential fatty acids [177]. Variations in the color and sensory characteristics of ruminant meat by dietary means can be mediated through the effect of production systems on growth rate, carcass weight at slaughter, carcass fatness, and intramuscular fat content and composition [178,179,180,181]. It is well known that the stability of muscle or meat to oxidation is the result of the balance between prooxidants and antioxidants, and the composition of PUFAs, proteins, pigments, and cholesterol substrates in the muscles [9,12,17]. The role of n-3 fatty acids in the space of offering antioxidative properties and improved product quality cannot be underestimated. The impact of n-3 FAs on lipid oxidation and color and flavor deterioration requires closer investigation because n-3 FAs may exert antioxidative activity in tissue systems and therefore may not detrimentally affect the flavor and color of meat unless the antioxidant capacity of the muscle system is compromised or lower than the threshold [182]. Similar to other PUFAs, the longer-chained highly unsaturated n-3 FAs in fish oil can be easily peroxidized to form hydroxyperoxide, which can increase oxidative stress in animals that consume such foods. However, research has shown that, in contrast to n-6 FAs, n-3 FAs in the muscle membrane systems are inhibitors of free radical generation [183]. Takahashi et al. [184] showed that diets rich in fish oil increased the expression of antioxidant genes in mouse liver and upregulated the expression of lipid catabolism genes. In addition, n-3 FAs have been shown to inhibit nitric oxide synthase expression and inducible nitric oxide synthesis by cytokine-activated macrophages [185]. A study involving goats showed that the inclusion of canola oil in the diet increased blood, muscle, and liver n-3 FA concentrations, but lipid peroxidation, as assessed by the concentration of TBARS, was significantly lowered in blood and muscle [186]. Together, these findings indicate that long chain n-3 FAs in the body can exert beneficial effects on the maintenance of welfare and the prevention of many diseases through the inhibition of free radical formation, most likely through cell signal transduction, gene expression, and/or antioxidant defense. However, one should be aware that vegetable oils are often naturally enriched with antioxidants such as vitamin E. The design of studies comparing fatty acid composition to antioxidant activity and immunity should account for differences in the vitamin E content in the fat source.

A recent study showed that a feedlot diet high in grain resulted in increased oxidative stress in lambs, as assessed by the blood biomarker isoprostane, and there was a positive correlation identified between muscle n-6 FA deposition (LA and AA) and blood isoprostane concentration [9]. It was observed that lipid oxidation of meat stored under simulated retail display for 72 h was linearly and negatively related to blood isoprostane concentration, but color deterioration assessed by redness of meat (*a** value) was not associated with blood isoprostane concentration. Muscle vitamin E content showed a significant negative linear relationship, but n-6 PUFA showed a significant positive linear relationship with lipid oxidation, respectively. Furthermore, it was found that increased muscle ALA concentration was positively related to the antioxidant enzymes of GPX_1_ and SOD_2_ in longissimus muscle, indicative of increased ALA in muscle tissues reducing oxidative stress in the animals [34].

Several studies conducted in animals have demonstrated that diets rich in n-3 PUFA can exert anti-inflammatory and immune modulatory effects in vivo with modulating cytokine production, lymphocyte proliferation, surface molecule expression, phagocytosis, and apoptosis, and inhibition natural killer cell activity [187,188]. The n-6 PUFA, especially AA and its precursors, enhance the production of the eicosanoid group of inflammatory mediators, such as prostaglandins, leukotrienes, and related metabolites. Instead, n-3 PUFAs act as antagonists of AA. Research into the effects of fatty acids on the immune system has revealed the roles of eicosanoids derived from AA in promoting inflammation; but the knowledge on long chain n-3 PUFA suggests that it can inhibit the metabolism of AA to produce its inflammatory mediators [189]. Studies conducted in laboratory animals showed an increased incorporation of n-3 fatty acids, as EPA and DHA, into phospholipids compared with saturated or monounsaturated fatty acids [190]. Feeding diets containing EPA can results in partial replacement of AA in the membrane of cells that are involved in inflammation, such as monocytes, macrophages, and neutrophils. This is the result of the decreased production of AA derived mediators, competition for cyclooxygenase and lipoxygenase enzymes, and decreased expression of cyclooxygenase-2 and 5- lipoxygenase [50].

In vitro and in vivo studies show that high dietary intake of omega-6 fatty acids (e.g., linoleic acid), usually sourced from corn, safflower, soybean, and sunflower oils, can partially inhibit lymphocyte proliferation, production of IL-2, CTL activity, NK cell activity, and the production of IgG and IgM [191]. Whereas n-3 PUFA, usually obtained from green leafy vegetables, rapeseed, and flaxseed, it can reverse the effects of PGE2, and simply act as a PGE2 antagonist. The excess PGE2 in the circulatory system can increase the production of pro-inflammatory cytokines (TNF, IL-1, and IL-6), increase the production of Th1-type cytokines, enhance MHCII expression, lymphocyte proliferation, and NK cell activity, and decrease IgE production by B lymphocytes [192]. ALA, an omega-3 FA, is a precursor for the formation of series-3 eicosanoids, including prostaglandins and eicosanoids, and are molecular signals associated with several functions in the body, including inflammation. Series 3 prostaglandins are anti-inflammatory, whereas series 2 prostaglandins, synthesized from AA, are pro-inflammatory [193]. A recent study reported that flushing in the late gestation period with supplementation of 2.8% α-linolenic acid to Ettawa Grade does improve the immune system postpartum [194]. These in vitro and in vivo studies associated with n-3 fatty acids supplementation suggest that dietary n-3 fatty acids can improve immune status by reducing inflammatory mediators indicative of antioxidant defense in the body. This may lead to improved health, performance [195], and quality of meat and milk produced in farm animals.

In the context of vitamin E as a dietary antioxidant, and taking beef cattle as an example, it offers a major contribution to the oxidative stability of beef. Dietary vitamin E is available from rangeland pasture grazing, whereas it is supplemented as vitamin premix in the feedlot rations used in intensive animal production. Native rangeland and improved pasture grazing systems offer substantial amounts of vitamin E (α-tocopherol), although the stage of plant maturity will impact concentrations. For example, animal feeds mainly made from dried grass hay or legume hay may offer very low concentrations of vitamin E. Cattle fed concentrate-mixed rations containing cereal grains and roughage in large proportions must instead source vitamin E from supplementation to achieve the recommended daily intake of 15–40 mg per kg dry matter [196]. It has been recommended that a vitamin E concentration of 3.3 mg per kg of beef is sufficient under commercial conditions for better retail display [14], although levels down to 0.9 mg per kg meat have also been reported as sufficient for short-term chilled storage durations. This is because, in the muscle tissue system of ruminants, vitamin E acts as a free radical scavenger and preserves fatty acids and proteins against oxidative deterioration and thereby protecting against myoglobin oxidation and color deterioration [15,18,197].

It is also well documented when antioxidant substrates in the muscle tissues are low or below the threshold, lipid oxidation is a serious concern in cooked meat. Specifically, the oxidation of lipids can occur at a higher rate in cooked meat as compared to raw meat due to the reaction of PUFA with light, oxygen, and high cooking temperatures. With regard to turkey, fatty acids in muscles are mainly phospholipid-bound and with a high level of unsaturated fatty acids, whereby the risk of lipid oxidation in the muscle meat system is even higher than other livestock species, including broiler chicken. Lipid peroxidation can produce free radicals, which cause release of heme and oxygen and ultimately produce off-flavors, which are more prevalent after repeated cooking of lamb meat [198]. In general, sheep meat contains high levels of fat and particularly n-3 PUFA, which is more susceptible to oxidation, compared to beef or pork [199]. Lipid oxidation is induced by unsaturated fatty acids and oxygen, which are initiators in the production of lipid hydroxides. Due to their unstable status, they can further decompose into secondary oxidation products and develop alkyl radical, peroxyl radical, lipid hydroperoxides, and alkoxyl radical and these may cause rancid odors and flavors in cooked meat.

There are many ways to control lipid oxidation in meat by adding antioxidants. For example, synthetic antioxidants such as butylated hydroxyl toluene (BHT) and butylated hydroxy anisole [145] at low levels are used in the food industry to control lipid oxidation and increase the shelf-life of processed meat products [200,201]. However, in the last decade, the application of natural antioxidants in food preservation is preferable to human food and animal feed regulatory authorities. This is to ensure food safety and address human health concerns, as some synthetic antioxidants or the application of nitrite in meat products are believed to increase the carcinogenic or mutagenic risk to consumers upon higher consumption [46,202]. Therefore, emphasis on the use of natural antioxidants has been recommendable [203,204]. Alternatively, meat can be combined with ascorbic acid, wrap flour, or antioxidants extracts from plants before cooking to prevent the formation of lipid oxidative products of hydroperoxides, carbonyl compounds, and epoxides [183]. Adding α-tocopherol is the most commonly used method either alone or in combination with ascorbic acid in food preparation and preservation [205,206]. However, other elements having antioxidant potential, such as selenium, magnesium, or zinc, as well as antioxidant rich plant extracts such as polyphenols. The efficacy of this latter element is not as well proven as compared to the application of vitamins as antioxidants [207].

### 6.3. The Impact of Dietary Antioxidants on Performance and Milk Productivity of Farm Animals

Under normal conditions, the intracellular levels of ROS are maintained at low to moderate levels by various enzyme systems that participate in the in vivo redox homeostasis and allow for better physiological conditions and health. Production of ROS at low to moderate levels can help the individual to cope with the pathogenic microorganisms, potentially via its involvement in regulating inter-microbial competition [208]. However, the production of various ROS in excess of endogenous antioxidant defense mechanisms promotes the development of a state of oxidative stress, with significant biological consequences. There has been evidence that oxidative stress plays a crucial role in the development and perpetuation of inflammation. It contributed to the pathophysiology of a number of debilitating diseases in humans [209] and caused a number of infectious diseases in farm animals [210].

During pregnancy, ROS exert a biphasic effect because adequate ROS concentration is essential for embryo development, implant, fetal defense against uterine infections, steroidogenesis, pregnancy maintenance, and partum. During physiological pregnancy, all tissues and, mostly, placenta and fetus require high amounts of oxygen. ROS, generated both by mother and fetus, are implicated in fetal growth because they promote replication, differentiation, and maturation of cells and organs. On the other hand, it is well known that over production of ROS may lead to oxidative stress and represent the underlying cause of many diseases in humans and animals [211]. It enforces the importance of balances between the oxidant substances and antioxidant defenses in biological systems of living organisms for healthy life.

In gravid ewes, based on the pregnancy rate (twin or triplets), the oxidative substance MDA increases proportionally in the circulating lipids, to the number of fetuses. It is likely, an increased antioxidant potential could counteract the effects deriving from higher ROS generation. Under this condition, the activity of antioxidant enzymes (SOD, GSHPx, GSH) and of related substances, such as selenium (Se), is deeply compromised due to the metabolic changes with high number of fetuses. Therefore, the antioxidant potential is found to decrease as the number of fetuses increases in the uterus [212]. Another reason for the reduction in antioxidant defenses could rely on the high lipid content found in maternal blood, during twin pregnancies [213]. In this context, the increased lipids in the blood might be utilized as energy substrates for the development of the fetus and deposition of fat in the mammary tissues for the post-partum milk production, which is associated with the survival of offspring. At lambing, O_2_ consumption increases to tripling level, resulting from the activation of the oxidase acting on NADPH, represents a true ‘respiratory explosion’, generated into phagocytes and leading to ROS generation, mostly as O_2_^−^ and H_2_O_2_ [214]. Lambing is in fact considered an inflammatory process, during which the uterus and cervix appear crowded by inflammatory cells, particularly neutrophils and monocytes. This leucocyte migration is triggered by an increase in cytokine generation, among which are interleukin 1, 6, 8 (IL-1, IL-6, IL-8), TNFα, and other chemotactic factors, in the uterus and cervix [215]. These cytokines promote the activation of metalloproteinases 1 and 8 (MP-1 and MP-8) and chemotaxis of local fibroblasts and leucocytes. These cells, in turn, generate some molecules, such as serotonin, prostaglandins, and vasoactive peptides, strongly implied in myometrial contractility and in cervical maturation, important for physiological lambing [22,216].

With regard to the environment, high heat and prolonged dry weather can enhance metabolic disorders and reduce immune function in ruminants [217]. The alteration in energy metabolism associated with increased energy expenditure and reduced feed intake in animals facing prolonged hot seasons can lead to heat induced oxidative stress, loss of body weight, or loss of milk production. In extreme cases, it can induce subclinical diseases, such as ketosis or liver lipidosis, due to intense mobilization of body fat reserves and negative energy balance in dairy animals such as cattle or goats [218]. Stress due to high heat exposure during late spring to early autumn seasons can impact milk yield and milk composition in dairy cattle. The reduction in milk is partially due to reduced dry matter intake (35~50%) and partially due to alteration in more chronic physiological and metabolic activities [219]. Dairy cattle showed improved lactation performance and reduced inflammatory responses, when rumen-protected methionine was added to the diet of cows during the transition period [220].

In the blood and tissues of animals, oxidative stress is driven by the imbalance between the production of ROS and the neutralizing capacity of antioxidant mechanisms. Some of the well-established antioxidants include glutathione, superoxide dismutase (SOD), and vitamins A and E have the capacity to lower oxidative stress. Osorio et al. [221] observed that methionine supplemented dairy cows had greater overall plasma oxygen radical absorbance capacity and greater concentrations of glutathione and carnitine in liver when compared with a control group. The authors indicated that supplemental methionine enhanced de novo glutathione and carnitine synthesis in liver and, thus, increased antioxidant and β-oxidation capacity. The greater decrease of IL-6 after calving coupled with lower plasma ceruloplasmin and serum amyloid A (SAA) in methionine supplemented cows indicated a reduction in proinflammatory signaling within liver.

### 6.4. The Impact of Dietary Antioxidants on Functional Aspects of Milk and Milk Products from Farm Animals

The accumulation of ROS in the body and development of oxidative stress is an important area in medical and nutritional research [222] as it can impact health and performance of humans and animals. In the case of farm animals, it can adversely affect the quality and preservative aspects of milk and their processed products to the point of consumption. Oxidative stress also can be caused by frequent and extreme changes in the climate. Previous research has shown that exposure to environments with an elevated environmental temperature and increased humidity decreased milk protein and milk casein concentration in lactating dairy cows [223,224]. These alterations in milk composition are likely to be, partly, due to the reduction in DMI associated with heat stress. This might have been also caused by heat stress resulting in reduced delivery of protein precursors to the mammary gland, and increased utilization of amino acids for biochemical processes (e.g., gluconeogenesis) [225]. Rumen-protected methionine feeding supported milk protein concentration, when compared with cows in the control group, had decreased milk protein concentration and milk casein concentration, partly due to the direct effect of methionine supplementation on protein synthesis pathways in mammary epithelial tissue [226].

Antioxidant supplementation in the diet can reduce free radical formation and the number of somatic cells in milk, which contributes to a lessening of milk waste in the supply chain. Dietary supplementation, such as vitamin E, vitamin C, carotene, and trace elements, such as Se, zinc or β-flavonoids, vitamin A, and manganese, have been proven useful to reduce the occurrence of udder infections and improve the quality of milk production, in terms of fat, protein, and somatic cell count [7,8]. It also improves the nutrient value of milk by adding value as a source of antioxidants for human consumption, with beneficial effects in the gastrointestinal tract and other tissues [227,228,229]. For example, functional attributes of GSH-Px and Se compounds (e.g., selenoproteins) present in milk can prevent lipid oxidation as well as act as antioxidants in human tissues or the body upon consumption of milk [230]. Others have reported that dietary Se supplementation to cows increased the concentrations of the oligoelement in milk, resulting in an increased intake of selenium by humans consuming dairy products [8,231]. The latter studies reported that milk Se concentrations were twice as high when Se yeast was fed to the cow compared with selenite or selenate as dietary supplements. Others have reported that Se supplementation to sows increased the concentrations of Se in sows colostrum and milk (for review see Surai and Fisinin [131]).

Weiss [232] suggested that the definition of high-quality milk must be rated on antioxidants present in milk because the quality of milk can also be based on the amount of antioxidants that it contains, improving the shelf-life of milk by reducing oxidation. Milk and several fractions of milk products were found to have antioxidant properties. Major protein fractions present in human and bovine milk (casein and whey) and some milk protein-derived peptides have been reported to have antioxidant activity [233,234]. In particular, specific casein hydrolysates from bovine milk demonstrate antioxidant activity via radical scavenging properties in both aqueous and lipid model systems [235]. Casein phosphopeptides (CPPs) derived from tryptic digests of casein also have a strong affinity to sequester divalent metal ions, such as non-heme iron and copper, two potential food prooxidants under specific conditions [236]. Bovine milk casein-derived CPPs may therefore serve to remove prooxidative metal catalysts, such as iron from an environment of oxidizable lipids, to prevent lipid oxidation [237]. For example, whey and casein proteins can inhibit lipid peroxidation and peroxyl/superoxide radicals generation in milk and skimmed milk. Furthermore, casein inhibits peroxide and TBARS (thiobarbituric acid reactive substances) formation, an oxidative substance that forms in the cellular or circulatory systems when farm animals or humans are under oxidative stress. Whey protein also inhibits copper-catalyzed peroxides, TBARS formation, and O_2_ uptake for ROS generation in the biological systems. Lactoferrin can bind iron and inhibit Fe-induced lipid peroxidation. Hydrolysates from milk, fermented milk, casein, and whey were found to be antioxidative [11,235]. These examples show that several components found in milk are active in preventing lipid peroxidation and maintaining milk quality and also point to their potential usage as ingredients in foods to provide products for enhanced consumer health [230,238].

Depending on their nature, milk antioxidants can be divided into protein and non-protein compounds [239]. The examples of some protein antioxidants are casein, peptides, superoxide dismutase, catalase, and glutathione peroxidase. The non-protein antioxidants include vitamins C, E, and carotenoids. An off-flavor development in milk is still a quality issue, but on average, most milk will have a good flavor for up to 14 days of storage [238]. Oxidized flavor is described as cardboard-like, metallic, or tallowy and can develop over time because of improper storage and handling of the milk. In some cases, an oxidized flavor can be detected, in the milk, soon after milking. There are several factors that influence the oxidative stability of milk, which can vary considerably between individual cows. These effects can be caused either by fatty acid composition or low molecular weight antioxidants, such as uric acid, ascorbic acid, and tocopherol [231,232]. The antioxidant activity of dairy products has also been found in fermented milk and different studies have established the ability of lactic acid bacteria to release certain compounds with antioxidant activity during fermentation of the milk [11].

## 7. Future Perspectives

Oxidative stressor events are likely to increase in frequency, into the future, as a result of agroclimatic variability and its impact on thermal loads, nutrient availability of feeds, and animal management. There is an increased community awareness and prioritization of animal welfare that recognizes that efforts to limit oxidative stress will also help limit its associated negative effects on farm animal health (metabolic disorders and disease development). From a consumer perspective, those who identify as ‘health conscious and/or clean and green’ expect meat and milk products to promote their health and well-being—outcomes that are delivered with the consumption of foods rich in antioxidants, essential fatty acids, and quality protein. In addition, this latter market demand offers a pathway by which industry can extend the shelf-life and improve the quality of meat and milk products viz. improved antioxidant capacities. From these examples, it is observed that there is an imperative to address oxidative stress in farm animals and, by doing so, enhance the concentrations of antioxidants in their meat and milk products.

Dietary antioxidants are a viable means to maintain redox homeostasis in farm animals, as well as enhance the antioxidant concentrations of their meat and/or milk products. These observations are often secondary to the capacity of a novel forage, feedstuff, or agricultural by-product to meet the nutritional requirements of the farm animal. In this respect, the hierarchy of importance must transition to adopt more holistic priorities and approaches. Specifically, future animal feeds and supplements must be selected with reference to their antioxidant capacity, effect on animal welfare and product physiochemical properties, and nutritional value with a total ration. Practice must adhere to legislation and food safety standards that define limits for antioxidant or bioactive compound concentrations in animals and their meat and milk products. This may promote the adoption of ‘natural feeds’ with antioxidant properties as opposed to supplementation with synthetically sourced antioxidants. Furthermore, it is apparent that researchers have focused on only a few antioxidants (i.e., vitamin E, selenium, etc.). Future studies must seek to identify antioxidants that act synergistically or accumulatively with each other, such as phytonutrients and other bioactive compounds. It is of vital importance that these studies be in vivo and adhere to sound experimental and statistical principles, to provide confidence in the results and their practical value to livestock producers, retailers, and eventually consumers.

Knowledge of oxidative stress, its mechanisms within biological systems, and its interactions with antioxidants help to provide clarity when quantifying the activity and effectiveness of dietary interventions on redox homeostasis. Much of the current information is based on the study of humans and animal model species—a potential limitation when interpreting prooxidant and antioxidant interactions in other species, such as the various farm animals reared for meat and milk. Ruminants, because of their unique symbiosis with rumen microbiota, have an additional barrier when making inferences, based on monogastric research, on the effects of dietary antioxidants. It would be prudent, therefore, to undertake species specific research to identify the mechanisms and contribution of ROS/RNS to oxidative stress, animal health, and the potential actions of dietary antioxidants.

## 8. Conclusions

There is a substantial body of research investigating the effects of dietary antioxidants, endogenous antioxidants, and metal-binding proteins on biological systems. Many of these have used human and model species to study the interactions between antioxidants and prooxidants and their contribution to health, disease, and well-being biomarkers. From these, valuable insight into the mechanisms of oxidative stress and their contributions to farm animal performance and meat-milk quality, have been gained. Dietary antioxidants have, correspondingly, been identified as a viable means to preserve the redox homeostasis of farm animals that are susceptible or exposed to oxidative stressor events. It is important, however, to undertake in vivo animal studies that provide reproducible and reliable evidence for these mechanisms in farm animals. This is needed to support the continued use of dietary interventions, using forages, specialized feeds, and supplements that are rich in antioxidants, as a management strategy. Indeed, with the adoption of novel feeds and bioactive compounds, more holistic research is necessary to ensure farm animal welfare, producer profitability, and consumer satisfaction with meat and milk products.

## Figures and Tables

**Figure 1 animals-12-03279-f001:**
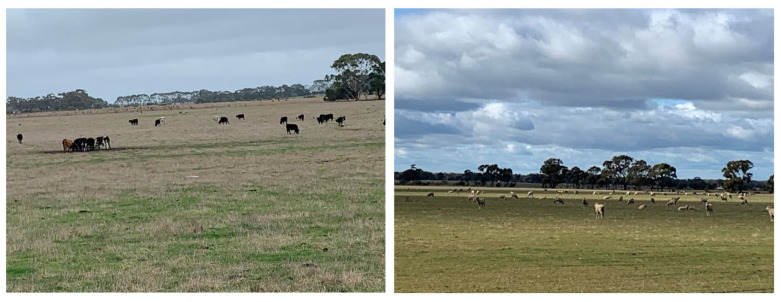
Feeding systems offer marginal amounts of antioxidants for daily maintenance and performance where the commencement of supplementary feeding with diets containing vitamins and minerals such as silage and/or a mix of grain is necessary for better health and productivity of animals. Cattle and sheep were grazing perennial ryegrass during autumn (April) 2021 and winter (July) 2022 seasons in the Western and Northern regions of Victoria, Australia.

**Figure 2 animals-12-03279-f002:**
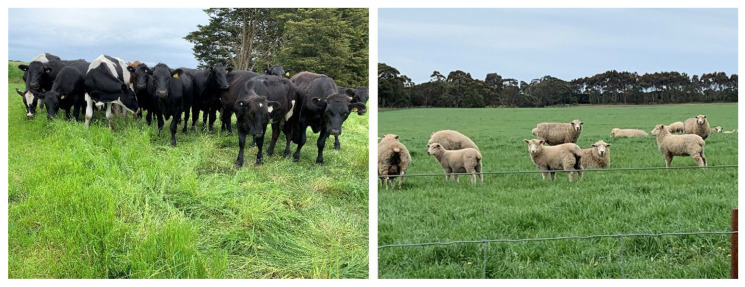
Feeding systems offer high antioxidants such as vitamins, minerals, and bioactives for daily maintenance and performance of farm animals, leading to better animal health, productivity, and milk and meat quality in cattle and sheep. Cattle and sheep (ewes and lambs) grazing solely on tall fescue (**left**) and perennial ryegrass (**right**) during spring in regional Victoria, Australia.

**Figure 3 animals-12-03279-f003:**
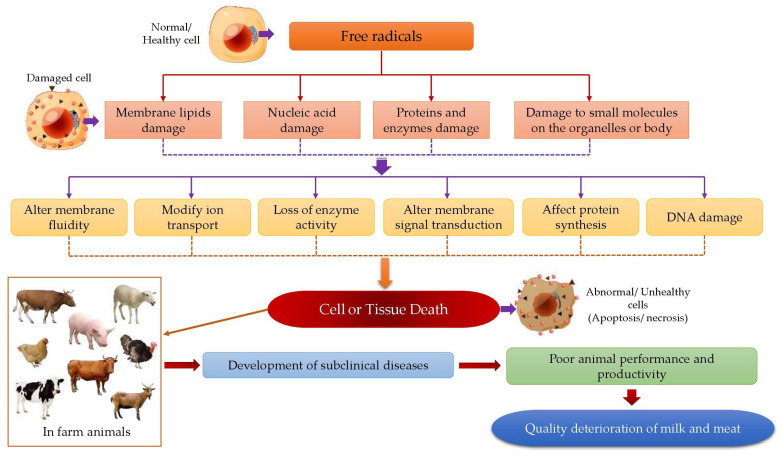
Impact of free radicals on the animal cells and tissues. Free radicals attack membrane lipids, proteins, nucleic acid, and small molecules on the cell and alter the key properties and functions of cells (membrane fluidity, ion transport, enzyme activity, membrane signal transduction, protein synthesis, and DNA damage). At extreme levels, cause cell or tissues necrosis, which result in the development of subclinical diseases, leading to poor animal performance, productivity, and consequently quality deterioration of milk and meat from farm animals.

**Figure 4 animals-12-03279-f004:**
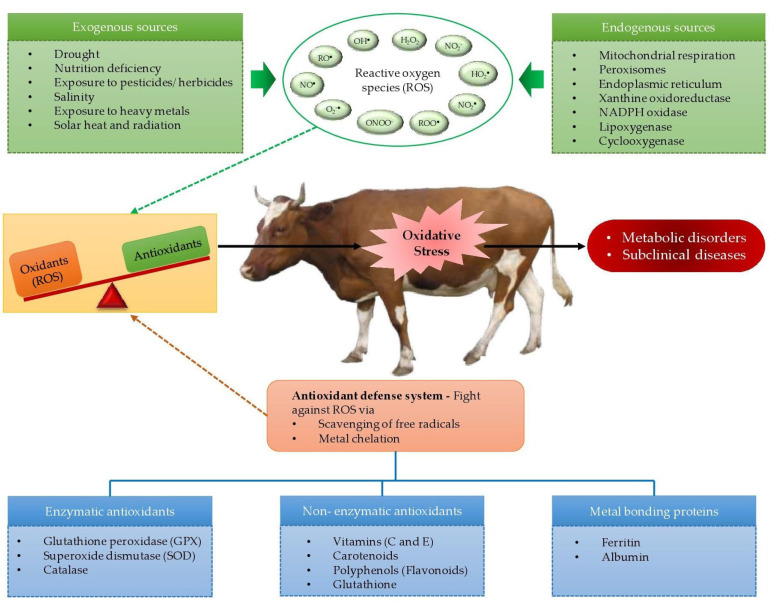
Development of oxidative stress in farm animals. ROS are generated by the induction of exogenous and endogenous sources. Enzymatic antioxidants, non-enzymatic antioxidants, and metal-binding proteins in the body fight against ROS via free radical scavenging and metal chelation mechanisms. When there is a disequilibrium between the ROS and antioxidants, ROS production will continuously increase and result in the development of a state called oxidative stress.

**Figure 5 animals-12-03279-f005:**
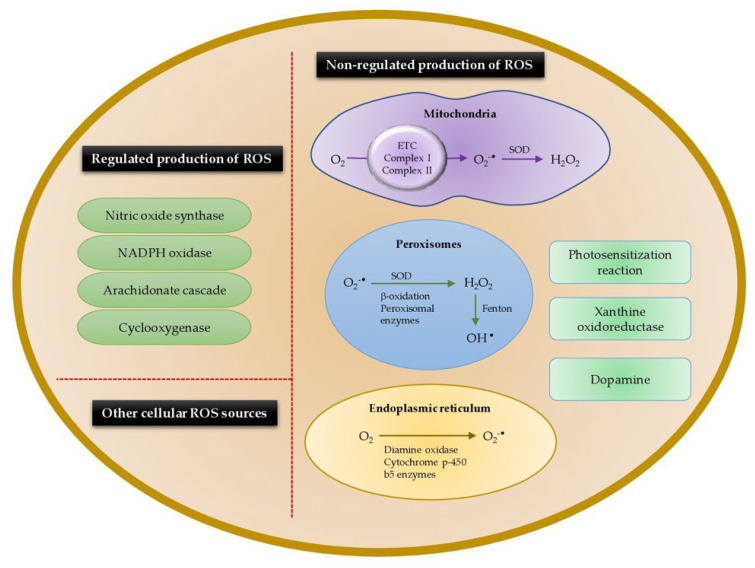
Sources of reactive oxygen species and free radicals in biological systems. The generation of ROS in the biological systems can be due to the non-regulated production of ROS, regulated production of ROS, and other cellular ROS sources.

**Figure 6 animals-12-03279-f006:**
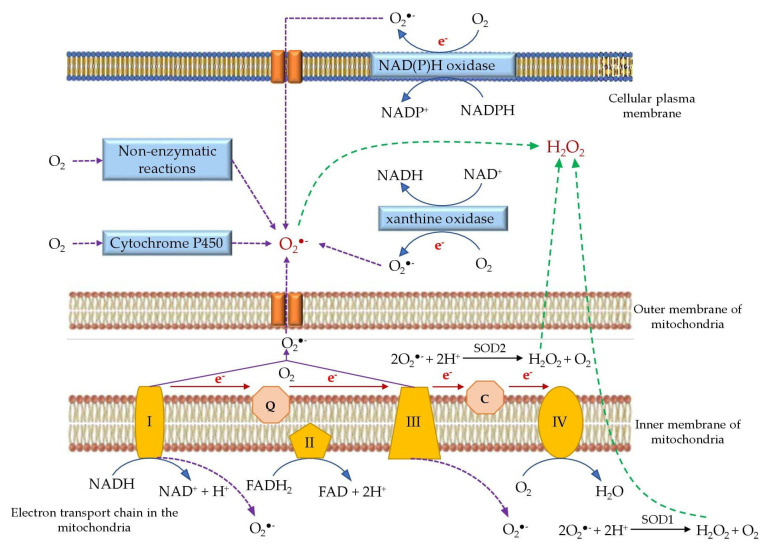
Generation of ROS in the mitochondria and cellular matrix of living organism. The superoxide radical (O_2_^•−^) is produced at a number of sites in the mitochondria, mainly complex I (sites IQ and IF) and complex III (site IIIQo). The produced superoxide radicals are converted into hydrogen peroxide by superoxide dismutase and transported to the cellular matrix. Besides, xanthine oxidase, Cytochrome P450, NAD(P)H oxidase, and non-enzymatic reactions also contribute to superoxide radical production in the cellular matrix.

**Figure 7 animals-12-03279-f007:**
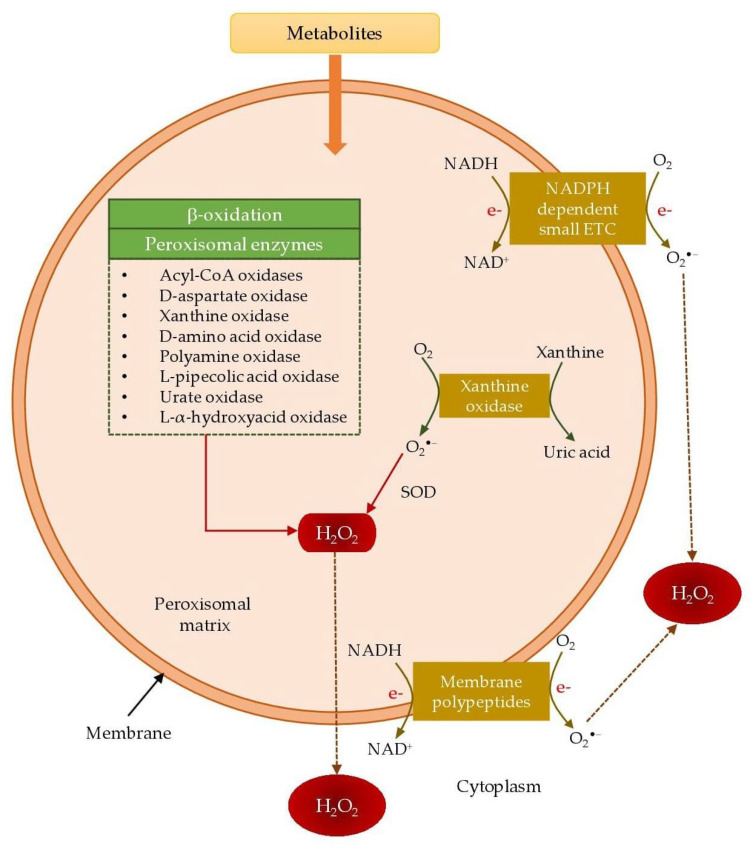
Process of H_2_O_2_ and O_2_^•−^ generation by the peroxisomes. The chief metabolic process that produces H_2_O_2_ in the peroxisome is the β-oxidation of fatty acids. Additionally, various peroxisome enzymes produce H_2_O_2_ as a part of their normal catalytic cycle. Moreover, NADPH dependent small ETC, xanthine oxidase, and membrane polypeptides produce O_2_^•−^ anions, which are further converted to H_2_O_2_.

**Figure 8 animals-12-03279-f008:**
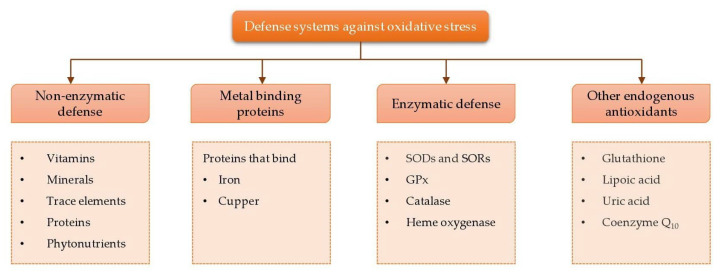
Types of defense systems against free radical formation in biological systems under stress conditions that affect performance, health, and well-being in human and farm animals. Defense systems can be categorized into enzymatic and non-enzymatic defense systems, metal binding proteins, and other endogenous antioxidants.

**Figure 9 animals-12-03279-f009:**
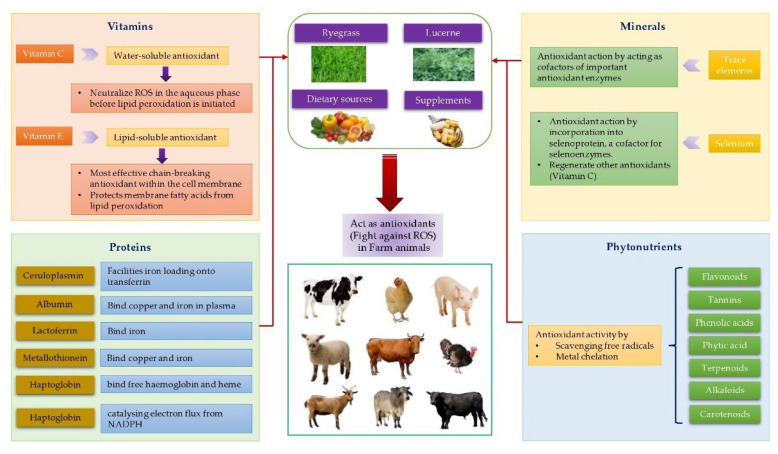
Non-enzymatic defense systems contributed by vitamins, minerals, proteins, and phytonutrients found in dietary sources. Different components from vitamins, minerals, proteins, and phytonutrients exhibit antioxidant activities in the biological systems by scavenging free radicals and/or chelating metal ions.

**Figure 10 animals-12-03279-f010:**
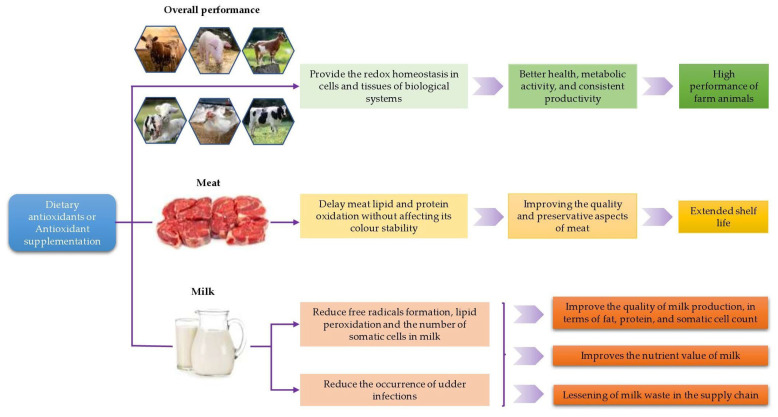
Impact of dietary antioxidants or antioxidants supplementation on the overall performance and meat and milk production of farm animals. The schematic diagram shows how diets rich in antioxidants can enhance animal productivity and improve quality and preservative aspects of meat and milk in farm animals.

**Table 1 animals-12-03279-t001:** Types of reactive oxygen species produced by several peroxisomal enzymes and their corresponding substrates.

Enzymes	Corresponding Substrates	Reactive Oxygen Species
Acyl-CoA oxidases	Long chain fatty acids, methyl branched chain fatty acids, bile acid intermediates	H_2_O_2_
D-aspartate oxidase	D-aspartate, *N*-methyl-D-aspartate	H_2_O_2_
Xanthine oxidase	Xanthine	H_2_O_2_, O_2_^•−^
D-amino acid oxidase	D-proline	H_2_O_2_
Polyamine oxidase	*N*-Acetyl spermine, spermidine	H_2_O_2_
L-pipecolic acid oxidase	L-pipecolic acid	H_2_O_2_
Urate oxidase	Uric acid	H_2_O_2_
L-α-hydroxyacid oxidase	Glycolate, lactate	H_2_O_2_

**Table 2 animals-12-03279-t002:** Production systems offering different status of antioxidant function and oxidative stress.

Cattle Production	Production Systems	Sheep Production
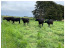	Grazing systems offering high to moderate antioxidant function (cattle and sheep grazing tall fescue pasture) (left) and perennial ryegrass (right) in spring.	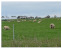
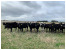	Grazing systems offering marginal antioxidants and point of ROS development and free radical formation. Photos taken in autumn season.	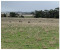
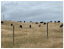	Poor antioxidant function and oxidative stress due to malnutrition caused by drought and lack of green grass availability. Photos taken in summer season.	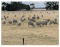
Incidence of high ketone bodies in the blood, increased somatic cells in milk, and dark cutting meat in cattle.	Initiation of cellular dysfunction and metabolic disorders, leading to reduced performance and product quality.	Lower muscle: fat ratio in the carcass and reduced eating quality of meat in sheep.

## Data Availability

Not applicable.

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
