# Peer review of "The Importance of Dietary Antioxidants on Oxidative Stress, Meat and Milk Production, and Their Preservative Aspects in Farm Animals: Antioxidant Action, Animal Health, and Product Quality—Invited Review"

_animals, 2022, doi:10.3390/ani12233279_

Round 1

Reviewer 1 Report

The manuscript is an extensive review on a very relevant problem concerning the importance of the antioxidants in the diet oxidative stress, meat and milk production, and their preservative aspects in farm animals with specific attention to  antioxidant action, animal health and product quality. The review provides the most recent insight on this very important problem. The review contains a total of 8 sections that describe in detail the mechanism of oxidation in the organism and go through the various types of antioxidants and their effect on the animal performance, productivity and product quality. The future perspectives are also very clearly outlined. The review also contains adequate number of tables and schemes/ figures. The reference list contains titles that are all into the context of the work and are properly sited. In my opinion the manuscript can proceed to publication.

Author Response

Dear Reviewer,

Thank you very much for your time and the valuable comments.

Kind regards,

Eric

Reviewer 2 Report

1- This  review is well designed and  covered all topics related to its aim.

2- In my opinion, LN 318 please add  Insulin-like growth factor I (IGF-I) 

3. LN 767: It is well known that antioxidant vitamins include vitamin E, vitamin B complex, and vitamin C.; Is it possible to add the benefits of vitamin b complex?

4- There a one suggestion may add values to this review. In my opinion, it will be great to add a paragraph discuss the relationship between  dietary antioxidants and reproductive performance and parameters (LN  1452) 

Author Response

Dear Reviewer,

Thanks for your time and valuable comments. We have answered your comments in the attached document and also have shown the additions on the revised version using track changes.

Kind regards,  Eric 

Reviewer 3 Report

I kindly accepted the invitation of review on the paper “The importance of dietary antioxidants on oxidative stress, meat and milk production, and their preservative aspects in farm animals: Antioxidant action, animal health and product quality – Invited Review”.

The above manuscript is an excellent work . The way of presenting the topic  is an interesting compilation of knowledge on this topic and can be treated as a model for this type of work. The authors' that is why they have my highest appreciation and therefore I am requesting that a given work be allowed to be published in the present form as it is presented.

I recommend the manuscript for publication in Animals. However the work needs small improvement according to the comments (file in attachment).

Kind regards,

Author Response

Dear Reviewer,

Thanks for your valuable comments. We have addressed your comments on the revised version using track changes.

Kind regards,

Eric